# Genome-Scale Mining of Acetogens of the Genus *Clostridium* Unveils Distinctive Traits in [FeFe]- and [NiFe]-Hydrogenase Content and Maturation

Pier Francesco Di Leonardo,[a] Giacomo Antonicelli,[a,b] Valeria Agostino,[a] Angela Re[a,c]

aCentre for Sustainable Future Technologies, Istituto Italiano di Tecnologia Foundation, Turin, Italy
bDepartment of Environment, Land and Infrastructure Engineering, Politecnico di Torino, Turin, Italy
cDepartment of Applied Science and Technology, Politecnico di Torino, Turin, Italy

Pier Francesco Di Leonardo and Angela Re contributed equally to this article. Author order was determined both alphabetically and in order of increasing seniority.

**ABSTRACT** Knowledge of the organizational and functional properties of hydrogen metabolism is pivotal to the construction of a framework supportive of a hydrogen-fueled low-carbon economy. Hydrogen metabolism relies on the mechanism of action of hydrogenases. In this study, we investigated the genomes of several industrially relevant acetogens of the genus *Clostridium* (*C. autoethanogenum*, *C. ljungdahlii*, *C. carboxidivorans*, *C. drakei*, *C. scatologenes*, *C. coskatii*, *C. ragsdalei*, *C.* sp. AWRP) to systematically identify their intriguingly diversified hydrogenases' repertoire. An entirely computational annotation pipeline unveiled common and strain-specific traits in the functional content of [NiFe]- and [FeFe]-hydrogenases. Hydrogenases were identified and categorized into functionally distinct classes by the combination of sequence homology, with respect to a database of curated nonredundant hydrogenases, with the analysis of sequence patterns characteristic of the mode of action of [FeFe]- and [NiFe]-hydrogenases. The inspection of the genes in the neighborhood of the catalytic subunits unveiled a wide agreement between their genomic arrangement and the gene organization templates previously developed for the predicted hydrogenase classes. Subunits' characterization of the identified hydrogenases allowed us to glean some insights on the redox cofactor-binding determinants in the diaphorase subunits of the electron-bifurcating [FeFe]-hydrogenases. Finally, the reliability of the inferred hydrogenases was corroborated by the punctual analysis of the maturation proteins necessary for the biosynthesis of [NiFe]- and [FeFe]-hydrogenases.

**IMPORTANCE** Mastering hydrogen metabolism can support a sustainable carbon-neutral economy. Of the many microorganisms metabolizing hydrogen, acetogens of the genus *Clostridium* are appealing, with some of them already in usage as industrial workhorses. Having provided detailed information on the hydrogenase content of an unprecedented number of clostridial acetogens at the gene level, our study represents a valuable knowledge base to deepen our understanding of hydrogenases' functional specificity and/or redundancy and to develop a large array of biotechnological processes. We also believe our study could serve as a basis for future strain-engineering approaches, acting at the hydrogenases' level or at the level of their maturation proteins. On the other side, the wealth of functional elements discussed in relation to the identified hydrogenases is worthy of further investigation by biochemical and structural studies to ultimately lead to the usage of these enzymes as valuable catalysts.

**KEYWORDS** *Clostridium*, acetogen, cofactor, electron bifurcation, hydrogenase

Address correspondence to Angela Re, angela.re@polito.it.

The authors declare no conflict of interest.

Hydrogen ($H_2$) metabolism is a widespread feature of microbial life, with many microorganisms using molecular hydrogen as a low-potential electron donor or generating it by reducing protons (1, 2). Exploiting the function of $H_2$ metabolism is pivotal in biological carbon sequestration, bioenergy, and bioremediation (3). $H_2$ stands out as the enabler of the lowest-cost low-carbon energy system, affording emission-free transport, heating, and industrial processes as well as interseasonal energy storage. In addition to being a fermentation end product, $H_2$ provides the energy source necessary for single carbon ($C_1$) gas refinery programs transforming abundant greenhouse gases such as carbon dioxide ($CO_2$) into value-added chemicals by a wide range of natural chemolithoautotrophic microorganims, including carboxydotrophic bacteria and acetogens (4–6).

To interconvert protons and electrons with molecular hydrogen, bacteria resort to two classes of metalloenzymes, [FeFe]-hydrogenase and [NiFe]-hydrogenase. This study does not take into account [Fe]-hydrogenases since they are specific of methanogenic archaea (1).

In this study, we carried out a thorough characterization of the hydrogenases' repertoire of acetogens of the genus *Clostridium*, which, owing to the efficient carbon fixation pathway, have already found commercial deployment to reduce CO and/or $CO_2$ using $H_2$ as an energy source and produce biofuels and biocommodities (7, 8). In spite of the prominent role of hydrogenases in acetogenic redox balance and energy metabolism as depicted in Fig. 1 (9), the activity of hydrogenases is often assayed globally (10–12), whereas a systematic study of the content of hydrogenases and their functionally relevant features is currently missing. The hydrogenases' repertoire has not yet been addressed systematically in clostridial acetogens, with few exceptions, such as the purification and characterization of the electron-bifurcating NADP- and ferredoxin-dependent [FeFe]-hydrogenase in *Clostridium autoethanogenum* grown on CO (13). Only some genetic outfits have so far been drafted on the basis of genomic annotations, such as those for *C. autoethanogenum* (14), *C.* sp. AWRP (15), *C. ljungdahlii* (16, 17), *C. carboxidivorans* (18), or *C. drakei* (19), or on the basis of knowledge transfer by presumed homology, such as for *C. ragsdalei* (10). To this aim, we relied on the availability of genome-scale amino acid sequence data of these microorganisms to develop an entirely *in silico* approach to systematically infer and bioinformatically validate and characterize their [NiFe]- and [FeFe]-hydrogenases' composition. Finally, we sought to identify, according to the current knowledge of sequence-based functional elements, the maturation proteins deemed necessary for [NiFe]- and [FeFe]-hydrogenases' biosynthesis. An in-depth analysis of the acetogenic sequences let us notice distinctive traits among the inspected acetogens, with just *C. scatologenes* and *C. drakei* featuring the complete gene sets encoding the maturation machinery of both the [NiFe]- and [FeFe]-hydrogenases.

## RESULTS

**Identification and functional classification of hydrogenases across clostridial acetogens.** We developed the scheme depicted in Fig. 2 to enable the prediction of hydrogenase enzymes and of their functional class by primary amino acid sequence alone. To this aim, we relied on the hydrogenase classification scheme predictive of biological functionality available at HydDB (20), which descends from the definition provided in references 1, 18, 21, and 22. Indeed, [FeFe]- and [NiFe]-hydrogenases are hierarchically classified into different groups and subgroups that differ from each other by biochemical features and functional role such as respiration, sensing, and fermentation. The HydDB classification scheme is based primarily on the topology of phylogenetic trees built from the amino acid sequence of hydrogenase catalytic subunits/domains, and it includes 29 subgroups (within 4 groups) of [NiFe]-hydrogenases and 6 subgroups (within 3 groups) of [FeFe]-hydrogenases. Our survey of hydrogenase genes relied on an *in silico* homology search, where the amino acid sequences of each strain were aligned to the controlled repository of hydrogenase catalytic subunits stored in HydDB (20). We then sorted the statistically significant hits of the query sequences and we leveraged their framing in the

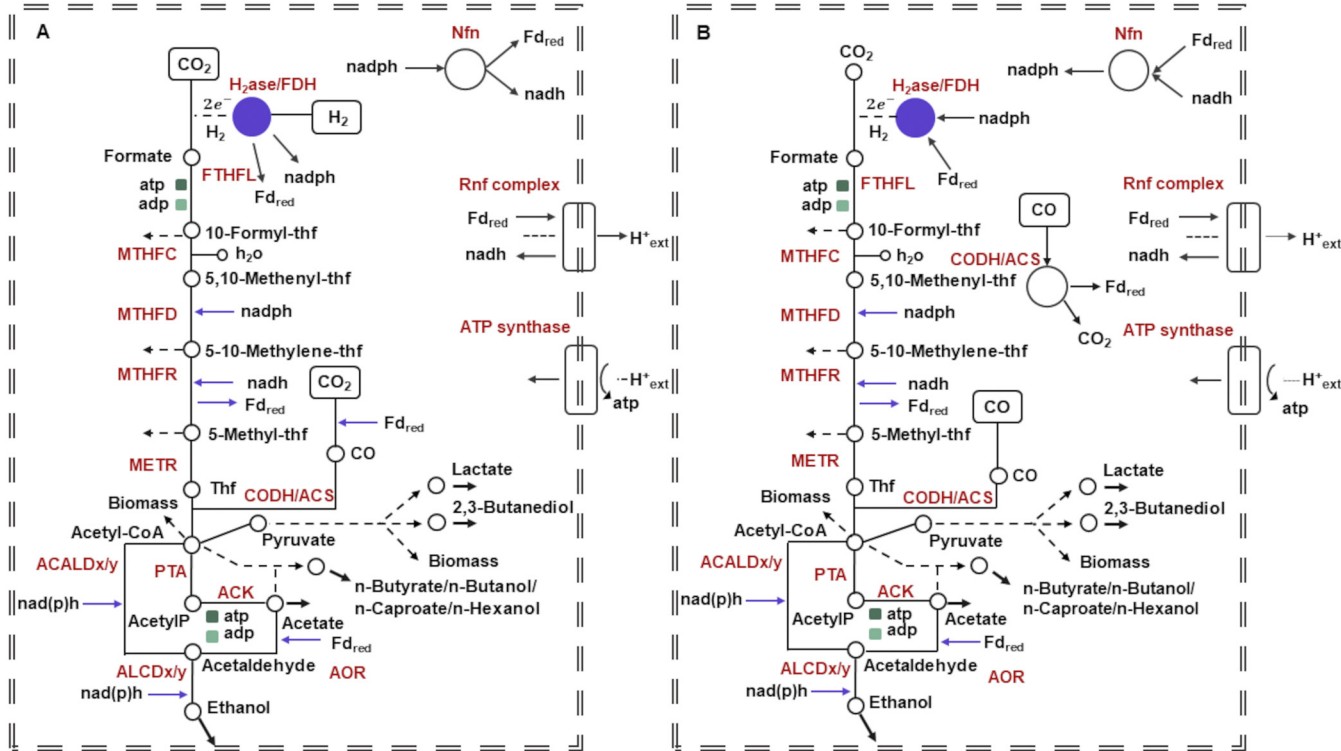

**FIG 1** Schematic overview of energy-conserving mechanisms in acetogens of the genus *Clostridium* without providing exact stoichiometries. (A) With $H_2$ as the electron source, the reducing equivalents for the reductive steps in the Wood-Ljungdahl pathway are provided by an $H_2$-oxidizing, electron-bifurcating hydrogenase/formate dehydrogenase complex (HytA-E/FDH) which reduces Fd, NADP, and $CO_2$. (B) With CO as the electron source, the reducing equivalents for the reductive steps are provided by the CO dehydrogenase/acetyl coenzyme A synthase (CODH/ACS), which reduces Fd. The hydrogenase protects the cells from overreduction when NADP and ferredoxin get too reduced during growth on CO. The electron-bifurcating and ferredoxin-dependent transhydrogenase Nfn is transferring electrons between Fd, NADH, and NADPH. The methylene-THF reductase is assumed to be electron bifurcating. Excess reduced ferredoxin ($Fd_{red}$) is oxidized by the Rnf complex, which reduces NAD and builds up an $H^+$ gradient. This gradient drives ATP synthesis via the $H^+$-dependent ATP synthase. Cofactors and energy equivalents are coded, respectively, in blue and green colors. Bold arrows denote byproducts' exchange reactions. Dashed arrowed lines denote reactions' sets that were collapsed for ease of reading. Metabolites are displayed in the figure. THF stands for tetrahydrofolate. Capital letters in red denote enzymes. FTHFL, formate:tetrahydrofolate ligase; MTHFC, methenyltetrahydrofolate cyclohydrolase; MTHFD, methylenetetrahydrofolate dehydrogenase; MTHFR, electron bifurcating; NAD-dependent electron-bifurcating methylenetetrahydrofolate reductase, METR, methyltetrahydrofolate:corrinoid methyltransferase; ACALDx/y, acetaldehyde:NAD(P) oxidoreductase; ALCDx/y, ethanol:NAD(P) oxidoreductase; PTA, phosphate acetyltransferase; ACK, acetate kinase; AOR, acetaldehyde:ferredoxin oxidoreductase.

HydDB classification scheme to predict the functional class of the putative hydrogenases. We subsequently carried out two checks to confirm if the amino acid sequences inferred by alignment-based sequence analysis were likely to encode hydrogenase catalytic subunits, as detailed in Materials and Methods. Supplemental file 1 reports the full list of the hydrogenase catalytic subunits identified by alignment-based sequence analysis along with the outcomes of the aforementioned checks. Of 444 catalytic subunits identified across the acetogens included in our study, 42 were deemed validated and were found to be distributed as shown in Fig. 3. The hydrogenases, which were ultimately discarded by effect of the checks carried out in addition to the preliminary hydrogenase identification, belonged to two main subgroups: group 4f [NiFe]- and group C2 [FeFe]-hydrogenases (supplemental file 1). Group 4f [NiFe]-hydrogenases are deemed to act as putative formate-coupled $H_2$-evolving [NiFe]-hydrogenases according to the HydDB classification scheme. The hypothetical nature of these hydrogenases was also highlighted in reference 23, which pointed out a couple of hypothetical group 4 [NiFe]-hydrogenases in *C. carboxidivorans* and *C. scatologenes*. Group C2 [FeFe]-hydrogenases may sense $H_2$ and regulate processes via methyl-accepting chemotaxis proteins, which are the most common receptors in bacteria and archaea (24). The genes predicted to encode hydrogenase enzymes upon completion of the validation procedure were found to be distributed into three major groups, group 1 [NiFe]-hydrogenase and groups B and A [FeFe]-hydrogenases, which separate into functionally distinct subgroups, as shown in Table 1. The soundness of

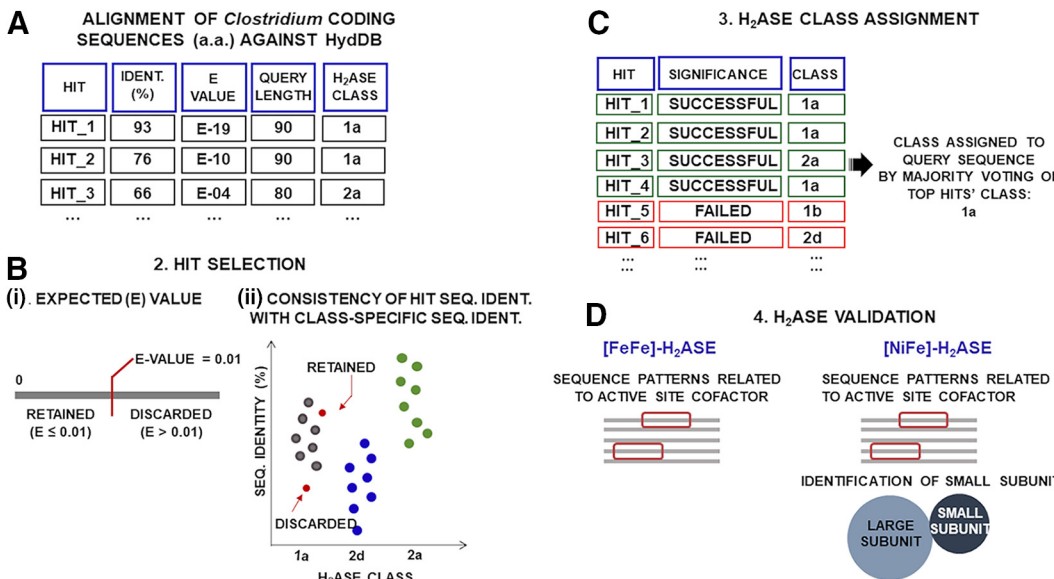

**FIG 2** Computational pipeline for the identification and validation of hydrogenase catalytic subunits. The computational workflow consists of several steps. (A) Alignment. Protein sequences of each acetogen were aligned against the controlled catalogue of [NiFe]- and [FeFe]-hydrogenases derived from HydDB. We sorted the hits in descending order of the following prioritized indices: percentage of sequence identity, E value, and query length. (B) Hit selection. Sorted hits were retained if the corresponding E value was ≤0.01 and if its percentage of sequence identity against the query met a class-specific criterion. Given the predicted class of the hit of a certain query sequence, we computed the distribution of the percentages of sequence identity between the hydrogenase sequences belonging to that class in HydDB. By way of example, we depicted the distributions corresponding to different classes in different colors in the figure. We retained an alignment hit if its percentage of identity against the query was higher than the minimum value of the aforementioned distribution. (C) Hydrogenase class assignment. We assigned the predicted hydrogenase to the class (group or subgroup, if necessary) shared by the majority of the four top hits. (D) Hydrogenase validation. We deemed a [FeFe]-hydrogenase catalytic subunit as validated if the protein sequence contained an amino acid motif known to coordinate the Fe-S cluster of the H-domain. For validating a [NiFe]-hydrogenase, we also explored the genes flanking each predicted [NiFe]-hydrogenase large catalytic subunit to ascertain the presence of a gene encoding a [NiFe]-hydrogenase small subunit. Abbreviations: H$_2$ase, hydrogenase; Seq. Ident., sequence identity; E value, expected value; a. a.: amino acid.

the hydrogenase class predictions is supported by the fact that the classification reflects the evolutionary relationships of [NiFe]- and [FeFe]-hydrogenases (see supplemental material, Fig. S1), in agreement with the phylogenetic distance of the acetogenic strains considered (25).

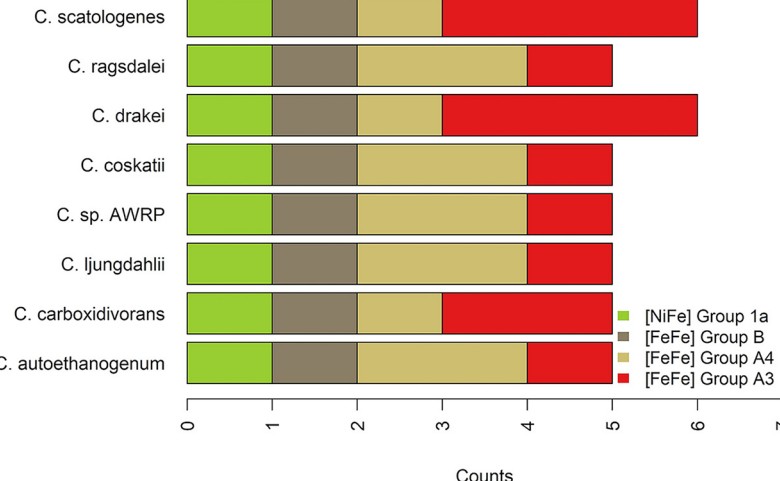

**FIG 3** Distribution of predicted [NiFe]- and [FeFe]-hydrogenase catalytic subunits in the acetogens of the genus *Clostridium* considered in this study. The bar plot displays the number of hydrogenases of each subgroup in each acetogen of the genus *Clostridium* examined in this study. Hydrogenase subgroups are color-coded.

**TABLE 1** Classification of [NiFe]- and [FeFe]-hydrogenase catalytic subunits in the acetogens of the genus *Clostridium* considered in this study

| Hydrogenase class | Class description | Subunit(s) of: | | | | | | | |
| --- | --- | --- | --- | --- | --- | --- | --- | --- | --- |
| | | *C. autoethanogenum* | *C. carboxidivorans* | *C. ljungdahlii* | *C.* sp. AWRP | *C. coskatii* | *C. drakei* | *C. ragsdalei* | *C. scatologenes* |
| NiFe Group 1a | Respiratory H2-uptake (periplasmic) | CLAU_RS04060 | Ccar_RS06390 | CLJU_RS14140 | DMR38_RS14545 | CLCOS_RS05460 | B9W14_RS11015 | CLRAG_RS17285 | Csca_RS07900 |
| FeFe Group B | Undetermined role. May couple $Fd_{red}$ oxidation to fermentative $H_2$ evolution | CLAU_RS00525 | Ccar_RS20210 | CLJU_RS09955 | DMR38_RS10360 | CLCOS_RS01500 | B9W14_RS24170 | CLRAG_RS12855 | Csca_RS20390 |
| FeFe Group A4 | Formate dehydrogenase-linked | CLAU_RS13700, CLAU_RS18770 | Ccar_RS16050 | CLJU_RS03480, CLJU_RS08485 | DMR38_RS03385, DMR38_RS08580 | CLCOS_RS06485, CLCOS_RS18190 | B9W14_RS20185 | CLRAG_RS09325, CLRAG_RS14505 | Csca_RS24095 |
| FeFe Group A3 | Bifurcating | CLAU_RS17440 | Ccar_RS01335, Ccar_RS10130 | CLJU_RS07205 | DMR38_RS07425 | CLCOS_RS00120 | B9W14_RS14620, B9W14_RS06375, B9W14_RS20215 | CLRAG_RS10115 | Csca_RS03805, Csca_RS12655, Csca_RS24060 |

**[NiFe]-hydrogenases in clostridial acetogens.** Each microorganism was found to encode a single $H_2$-uptake [NiFe]-hydrogenase enzyme belonging to the subgroup 1a, whose members primarily act as hydrogen oxidation catalysts, making hydrogen an additional usable inorganic electron donor for potentially generating energy. The small subunits of $H_2$-uptake [NiFe]-hydrogenases usually possess a large signal peptide containing a conserved R-R-x-F-x-K motif, which serves as signal recognition to target fully folded mature uptake hydrogenases to the membrane thanks to the recognition by a specific protein translocation pathway designated the membrane targeting and translocation (Mtt) (26) or twin arginine translocation (Tat) pathway (27, 28). No evidence for twin arginine signal peptides was gathered in any of the predicted [NiFe]-hydrogenase small subunits, as shown in the supplemental material.

We note that group 1a [NiFe]-hydrogenases are expected to be trimeric, but the analysis of the genes in the neighborhood of the large and small subunits detected a gene predicted to be a cytochrome $b_5$ only in *C. carboxidivorans*, *C. drakei*, and *C. scatologenes*. However, when we inspected the acetogenic genomes for the genes responsible for the cytochrome $b_5$ biosynthesis, we could identify putative candidates only for a few of the coproporphyrin-dependent (CPD) heme biosynthesis pathways by alignment-based sequence similarity (Table S3). The InterPro classification of the small subunits of the [NiFe]-hydrogenases in the remaining clostridial acetogens highlighted the presence of a domain featured by the C-terminal portion of cytochrome $c_3$ hydrogenases (IPR027394), as shown in supplemental file 1. However, cytochromes have not yet been characterized in any acetogen analyzed in this study (29). Furthermore, of the three pathways to assemble cytochrome c (30), prokaryotes can employ the pathways known as system 1 (31) or system II (32–34). However, with very limited exceptions, neither the system I nor the system II genes appeared to be carried in the acetogenic genomes according to our sequence similarity search (supplemental material, Table S4).

**[FeFe]-hydrogenases in clostridial acetogens.** The genome of each microorganism considered in this study contains a single gene encoding a group B [FeFe]-hydrogenase enzyme. A representative hydrogenase of this subgroup, CpIII in *C. pasteurianum*, has been recently biochemically and spectroscopically characterized in comparison to the group A2 CpI and to the group A3 CpII [FeFe]-hydrogenases (35, 36). As a result of this characterization, preferential stabilization of key catalytic intermediates through subtle changes in the outer coordination sphere was found to result in stabilization and/or destabilization of different oxidation states, and, in particular, CpIII was found to have a catalytic bias toward $H_2$ production (35, 36).

Each acetogenic genome was found to contain from one to three electron-bifurcating group A3 [FeFe]-hydrogenases. Whereas most acetogens were predicted to harbor a single group A3 [FeFe]-hydrogenase, two instances were identified in *C. carboxidivorans* and three instances in *C. scatologenes* and *C. drakei*. As shown in Fig. 4, prokaryotes use electron bifurcation to couple exergonic and endergonic chemical reactions from a two-electron donor to two widely separated one-electron acceptors in a single enzymatic complex (37).

Each acetogenic genome investigated in our study was predicted to encode from one (*C. carboxidivorans*, *C. drakei*, and *C. scatologenes*) to two (*C. ragsdalei*, *C. coskatii*, *C.* sp. AWRP, *C. ljungdahlii*, and *C. autoethanogenum*) group A4 [FeFe]-hydrogenases which are deemed to couple formate oxidation to $H_2$ evolution. One of the group A4 [FeFe]-hydrogenases that we predicted in *C. autoethanogenum* is supported by previously published data. Indeed, in 2013, a NADP-specific electron-bifurcating [FeFe]-hydrogenase consisting of six subunits $HytABCDE_1E_2$ was purified from *C. autoethanogenum* grown on CO and found to form a complex with a formate dehydrogenase (9, 13).

We believe that the predicted formate dehydrogenase-linked [FeFe]-hydrogenases could deserve future attempts to experimentally assess their physiological functions in the metabolism of the acetogens considered.

**Dissection of hydrogenase composition in individual clostridial acetogens.** We limit the discussion of the hydrogenases identified (Fig. 5) with the support of systematic cross-references to preacquired knowledge in the literature, whenever possible,

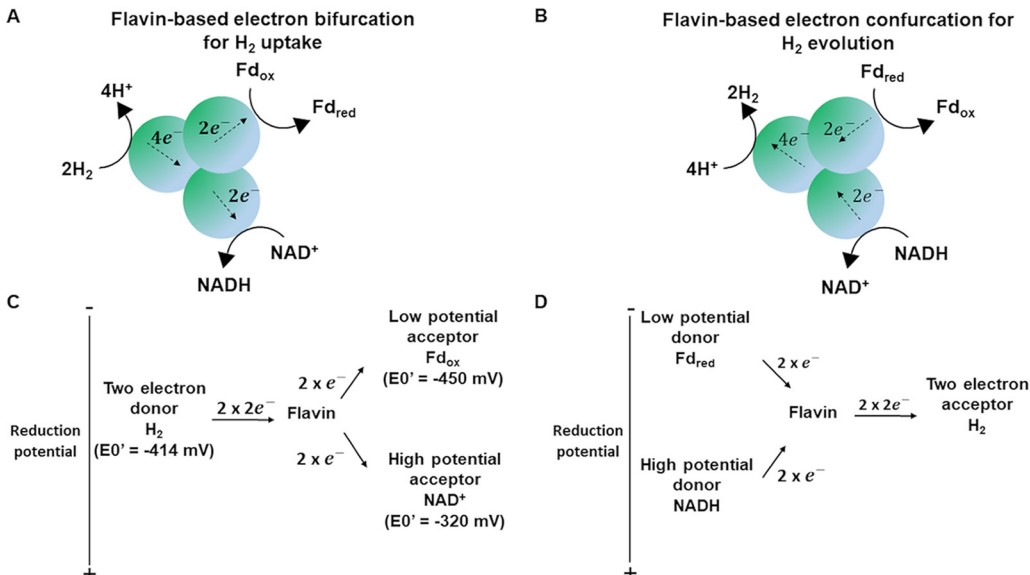

**FIG 4** Bifurcating/confurcating hydrogenases. (A) Flavin-based electron-bifurcating hydrogenase oxidizes an electron donor ($H_2$) and delivers the electrons simultaneously to two different electron acceptors. The reduction of a high-potential acceptor ($NAD^+$) is exergonic and drives the endergonic reduction of a low-potential acceptor (ferredoxin). (B) The flavin-based electron-confurcating hydrogenase simultaneously oxidizes NADH and ferredoxin to produce $H_2$. In this scenario, the electron pairs from the two electron donors, NADH and reduced ferredoxin, converge to reduce protons to $H_2$. Flavin is the cofactor family with bifurcating/confurcating property. (C) Conceptual illustration of ideal electron bifurcation whereby the transfer of electrons of intermediate reduction potential to a bifurcating site (flavin) is parsed out to acceptors with, respectively, more positive and more negative reduction potential but whose sum is equivalent to the overall reduction potential of the donated electrons. (D) A two-electron acceptor of intermediate reduction potential simultaneously accepts electrons from electron donors with, respectively, more negative and more positive reduction potentials.

only to *C. autoethanogenum*, for sake of brevity. Visual representation of hydrogenases' composition in each acetogen is available in Fig. 5 to 8. Full dissection of hydrogenases in each acetogen is available in the supplemental material.

The group B [FeFe]-hydrogenase encoded by CLAU_RS00525 corresponds to CAETHG_0110 reported in reference 9.

The group 1a [NiFe]-hydrogenase was predicted to consist of the large (CLAU_RS04060) and small (CLAU_RS04065) subunits. This enzyme has been previously mentioned in reference 9 as a dimeric [NiFe]-hydrogenase (CAETHG_0861-0862). According to the genetic organization of the blastp best hit (WP_010890826.1) of the catalytic subunit, which depicts the most common organization within this hydrogenase subgroup, we expected this enzyme to contain an accessory cytochrome subunit. Since variations often occur in the genetic organization within subgroups (22), we counted CLAU_RS04060 to RS04065 in the list of hydrogenases predicted for *C. autoethanogenum* even if we could not identify the accessory subunit.

The gene cluster CLAU_RS13680 to RS13705 (corresponding to CAETHG_2794 to 2799) encodes the group 4 [FeFe]-hydrogenase known as the electron-bifurcating NADP- and ferredoxin-dependent hydrogenase HytABCDE$_1$E$_2$. CLAU_RS13700 encodes the catalytic subunit HytA, while CLAU_RS13685 encodes the iron-sulfur flavoprotein HytB, which was predicted to harbor the site binding the redox cofactor. CLAU_RS13680, CLAU_RS13690, CLAU_RS13695, and CLAU_RS13705 encode the Fe-S subunits HytC, HytD, HytE$_1$, and HytE$_2$. An additional group A4 [FeFe]-hydrogenase was predicted to be encoded by CLAU_RS18770-RS18765 with CLAU_RS18770 acting as the catalytic subunit. According to the genetic organization of its best hit (WP_013238388.1) in the sequence similarity analysis against the HydDB database, we foresaw this enzyme to encompass two Fe-S cluster-containing subunits. However, owing to the uncharacterized state of the protein, we could not gather evidence from any of the databases employed to

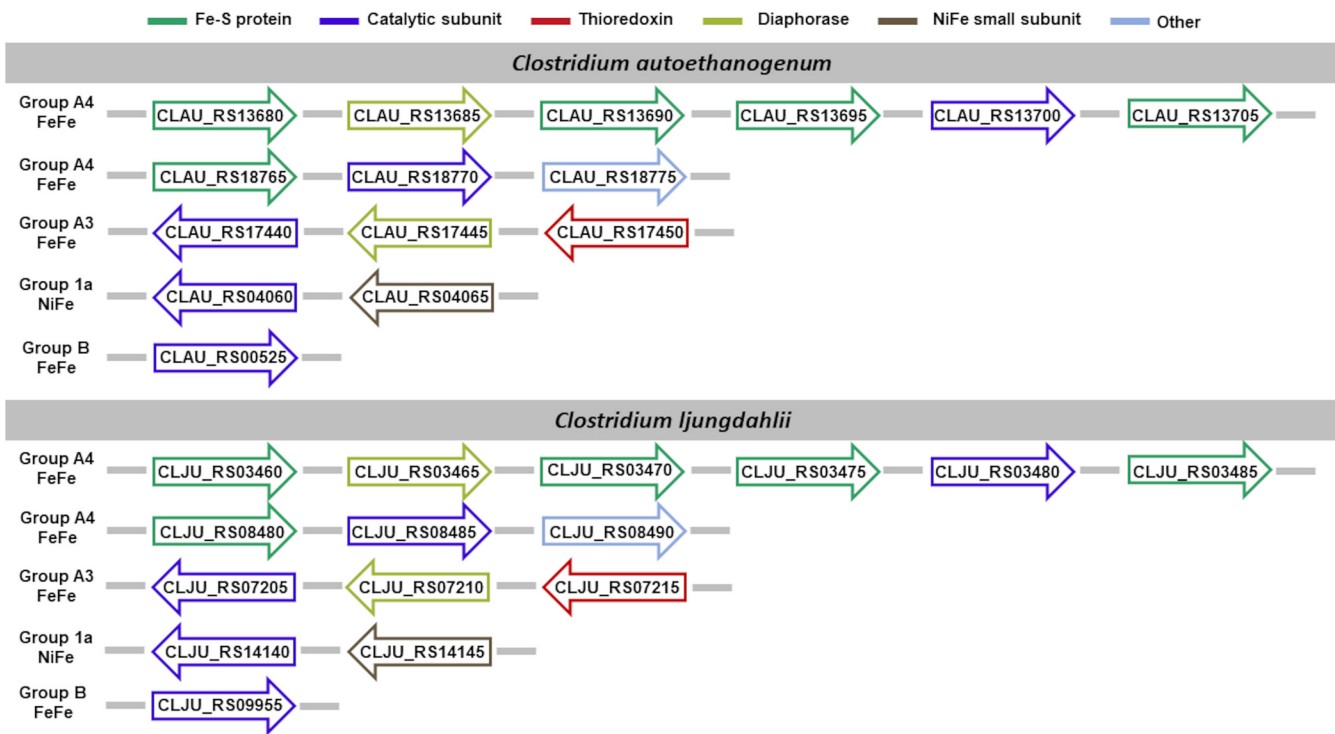

**FIG 5** Hydrogenase content in *C. autoethanogenum* and *C. ljungdahlii*. Shown is the genomic organization of the hydrogenase subunit-encoding genes. Subunits are color-coded based on their functional role, assigned in agreement with the neighboring genes' template typical of the hydrogenase class predicted by HydDB.

annotate proteins that the gene flanking the aforementioned catalytic subunit, CLAU_RS18775, encodes the second expected Fe-S protein.

According to our analysis, the genes CLAU_RS17440 to RS17450 encode a trimeric group A3 [FeFe]-hydrogenase which is predicted to operate in electron bifurcation mode. CLAU_RS17440 encodes the catalytic subunit, while CLAU_RS17445 encodes the redox cofactor-binding subunit and CLAU_RS17450 encodes a thioredoxin. The genes encoding this hydrogenase (CAETHG_3569 to 3571) were also identified in reference 9. This work reported an additional trimeric hydrogenase CAETHG_1576 to 1578 corresponding to CLAU_RS07605 to RS07615, which we did not include among the predicted hydrogenases, although it was present in the initially identified proteins according to sequence similarity analysis, since we could detect only two of the three P1, P2, and P3 metal-binding motifs. However, we think that CAETHG_1576 to 1578 deserves further investigation, especially in light of the effects of its inactivation on growth and product profiles reported for *C. autoethanogenum* grown on $H_2$-rich gas (38). Transcriptome analysis by transcriptome sequencing (RNA-seq) carried out in *C. autoethanogenum* cultures grown on syngas showed that the genes encoding the subunits of the hexameric group A4 [FeFe]-hydrogenase HytABCDE$_1$E$_2$ are by far the most highly expressed. They are followed by CLAU_RS00525, encoding the monomeric [FeFe]-hydrogenase, and by the genes CLAU_RS18765 to RS18775, which encode the trimeric group A4 [FeFe]-hydrogenase (Table 2).

**Hydrogenase maturation proteins.** In order to corroborate the identification of the hydrogenases in the clostridial acetogens investigated, we carried out a bioinformatic survey of proteins responsible for the maturation of [FeFe]- and [NiFe]-hydrogenases.

**[FeFe]-hydrogenase maturation proteins.** Maturation of [FeFe]-hydrogenases requires the biosynthesis and assembly of the peculiar H cluster consisting of a canonical [4Fe–4S]$_H$ cluster linked by a cysteine thiolate to a unique binuclear [2Fe]$_H$ cluster (39, 40). Each Fe center in the [2Fe]$_H$ cluster features terminal CO and CN$^-$ ligands, and the two Fe centers are bridged with CO and azadithiolate ligands, of which the

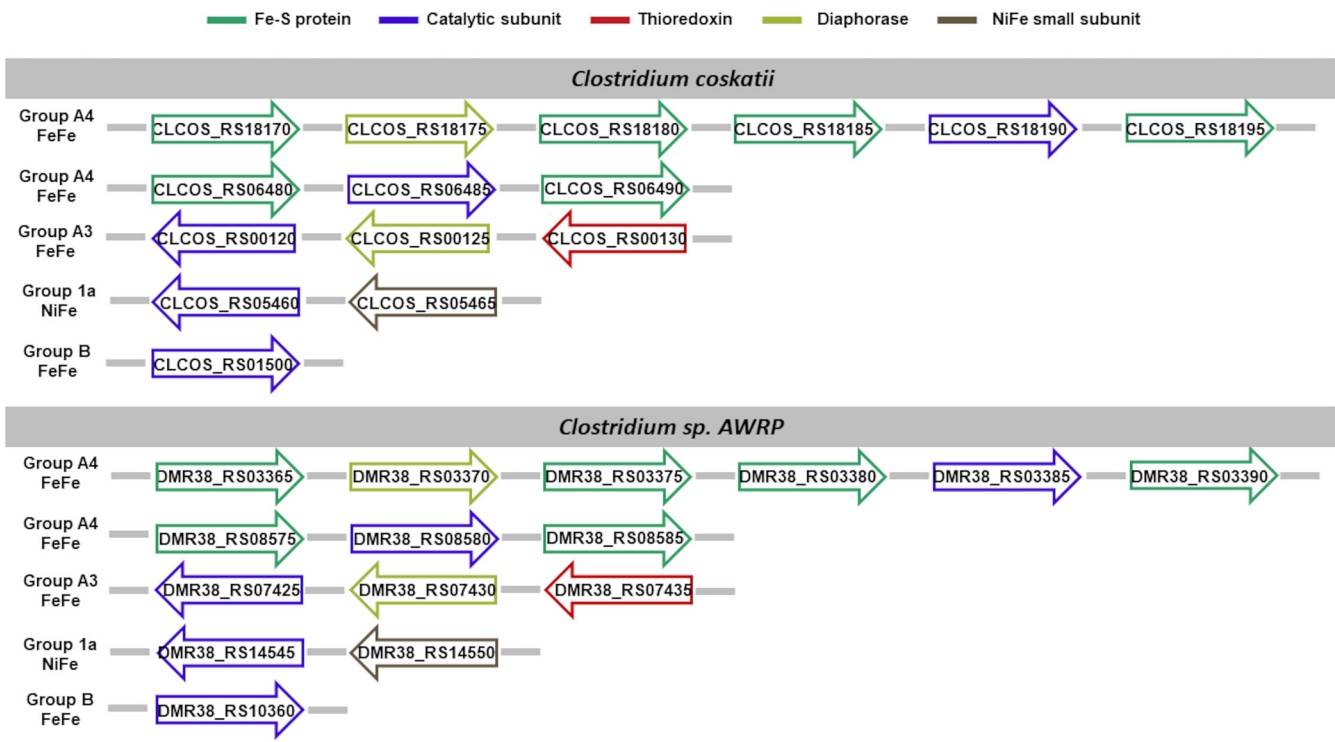

**FIG 6** Hydrogenase content in *C. coskatii* and *C.* sp. AWRP. Shown is the genomic organization of the hydrogenase subunit-encoding genes. Subunits are color-coded based on their functional role, assigned in agreement with the neighboring genes' template typical of the hydrogenase class predicted by HydDB.

secondary amine is thought to relay $H^+$ to/from a coordination site on the Fe that is distal to the $[4Fe–4S]_H$ cluster (41–43). The assembly of active [FeFe]-hydrogenases follows a complex pathway, where the $[4Fe–4S]_H$ cluster is installed by the regular Fe–S cluster assembly whereas the construction of the $[2Fe]_H$ cluster and its linkage to the $[4Fe–4S]_H$ involves three proteins HydE, HydF, and HydG (39, 44). The engagement of this set of proteins in [FeFe]-hydrogenases' maturation was initially identified in *C. reinhardtii* (45) and further investigated by heterologous expression of *C. reinhardtii* HydA1 and *C. saccharobutylicum* HydA with various combinations of HydE, HydF, and HydG in *Escherichia coli* (45, 46). These pioneering studies set the stage for the mechanistic characterization of the individual role of HydE, HydF, and HydG in the assembly of the $[2Fe]_H$ cluster. HydG is responsible, via a radical S-adenosylmethionine (SAM)-dependent interaction with its tyrosine substrate (47, 48), for forming both the CO (49) and $CN^-$ (50) ligands of the $[2Fe]_H$ subcluster, in the form of an organometallic Fe(CO)2CN synthon (39, 51) or as free diatomics (52–55). HydE is deemed the enzyme responsible for the synthesis of the dithiolate [2Fe]–bridging ligand synthesis, although both its substrate and its reaction mechanism are currently unresolved (54, 55). The GTPase HydF is a scaffold or carrier for the $[2Fe]_H$ cluster, prior to its insertion into the hydrogenase (39, 56).

**Radical SAM maturation proteins HydE and HydG in clostridial acetogens.** Both HydE and HydG belong to the radical SAM superfamily of proteins (57). Their radical SAM domains contain conserved motifs among which C-x(3)-C-x(2)-C (45) and C-x(2)-C-x (4)-C (54) are characteristic. Mutations of conserved Cys residues in the SAM-binding motifs of either *C. acetobutylicum* HydE or HydG resulted in defective maturation of the *C. acetobutylicum* [FeFe]-hydrogenase HydA (58). Furthermore, we found additional motifs typical of Fe-S cluster binding sites in the C-terminal ends of HydE and HydG (45, 57). The variable nature of the C-terminal motifs is notable since the amount of their sequence and/or functional conservation differs between HydE and HydG and even among HydG proteins (59). In HydE, the C-terminal Fe-S cluster is partially conserved and

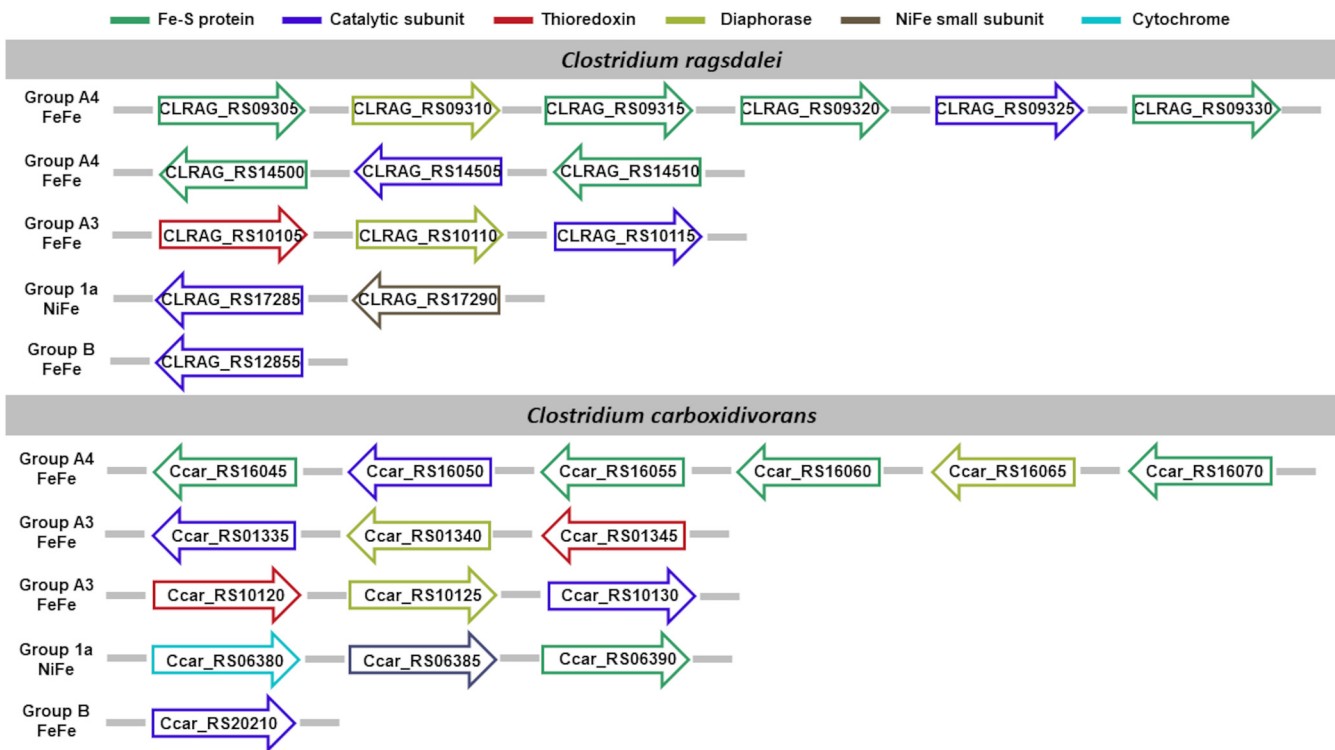

**FIG 7** Hydrogenase content in *C. ragsdalei* and *C. carboxidivorans*. Shown is the genomic organization of the hydrogenase subunit-encoding genes. Subunits are color-coded based on the functional role, assigned in agreement with the neighboring genes' template typical of the hydrogenase class predicted by HydDB.

can be coordinated by the conserved C-terminal C-x(7)-C-x(2)-C motif (57). However, in HydE, the presence or absence of this motif, and therefore of the cluster coordinated, does not affect the reaction mediated by this enzyme (54, 60). Instead, in HydG, the additional C-terminal Fe-S cluster, coordinated by the most known C-x(2)-C-x(22)-C motif (45, 52, 58, 61), was shown by mutational analysis to be essential for [FeFe]-hydrogenase maturation (58). However, the amino acid sequences responsible of coordinating this Fe-S cluster can be subject to variability (59).

When we inspected the acetogenic genomes for the radical SAM proteins HydE and HydG, by using the current knowledge from the literature, we were able to identify both of them in only three of eight *Clostridium* species, *C. autoethanogenum*, *C. drakei*, and *C. scatologenes*, as shown in Table 3. Table S7 reports the detailed reconstruction of motifs in the amino acid sequences. According to a previous [FeFe]-maturation protein annotation in *C. autoethanogenum* (9), CAETHG_0339 (CLAU_RS01585) encodes the HydG protein. Its amino acid sequence contains the radical SAM enzyme signature C-x(3)-C-x(2)-C but is missing the conserved Fe-S cluster binding motif on its C-terminal end. Here, instead, CLAU_RS03260 is deemed to plausibly encode the HydG protein in *C. autoethanogenum* since it harbors both signatures of a typical HydG protein, with the C-x(5)-C-x(19)-C motif supposed to coordinate the C-terminal Fe-S cluster. In the genome of the other five *Clostridium* species studied, we were not able to detect putative genes coding the HydG maturation protein.

**HydF maturation protein in clostridial acetogens.** HydF is a cation-activated GTPase, (58, 62), and its N-terminal end harbors amino acid sequence motifs, which are typical of this protein family and allow the interaction with GTP (63). The conserved G-x(4)-G-K-[S/T] P-loop is responsible for the binding of $\alpha$- and $\beta$- phosphate groups of the nucleotide. The h(4)-D-x(2)-G motif (h, hydrophobic) is responsible for the interaction with $\gamma$- phosphate and $Mg^{2+}$. The [N/T]-K-x-D interacts with the nucleotide. HydF is also an Fe-S protein whose C-terminal end presents the conserved C-x-H-x(46-53)-C-x(2)-C motif, which coordinates a [4Fe-4S] cluster essential for the hydrogenase maturation

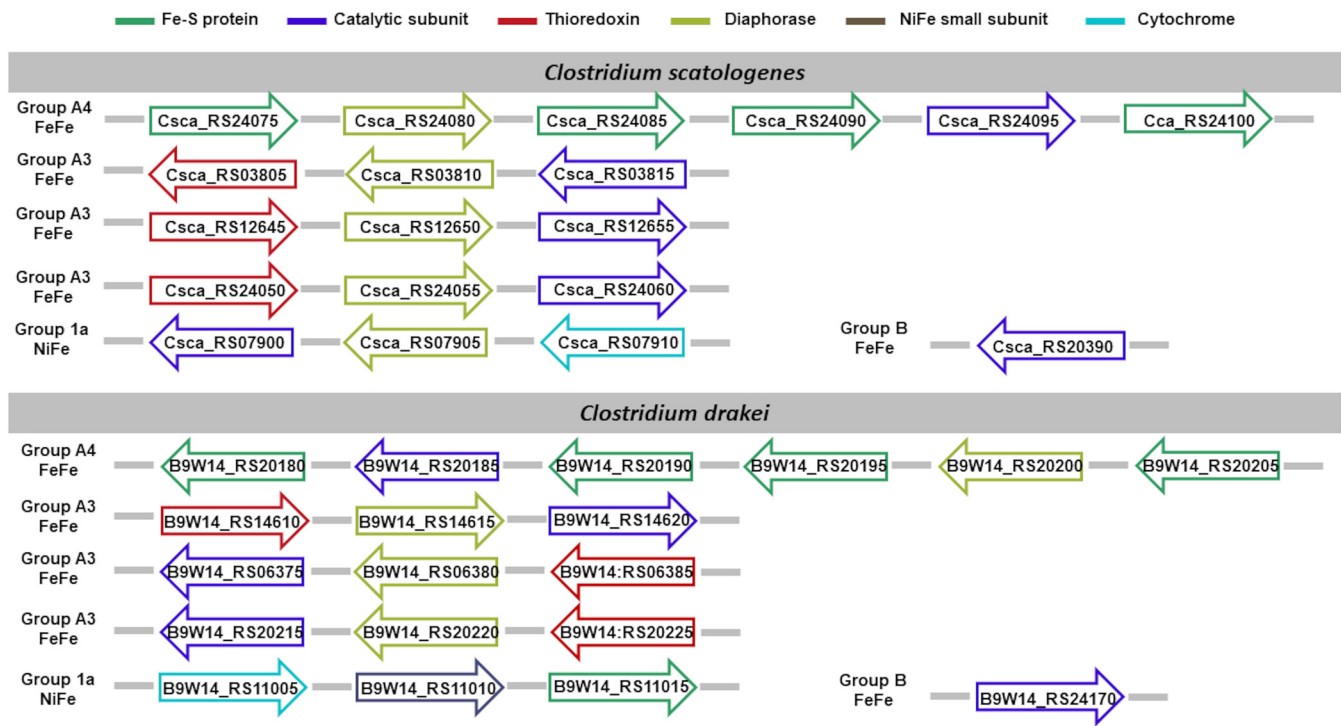

**FIG 8** Hydrogenase content in *C. scatologenes* and *C. drakei*. Shown is the genomic organization of the hydrogenase subunit-encoding genes. Subunits are color-coded based on the functional role, assigned in agreement with the neighboring genes' template typical of the hydrogenase class predicted by HydDB.

(58). Table 3 reports the genes encoding HydF proteins, and Table S9 reports the detailed reconstruction of the motifs featured by the amino acid sequences.

**[NiFe]-hydrogenase maturation proteins.** The maturation process of [NiFe]-hydrogenases is possible thanks to the primary action of six enzymes, known as HypABCDEF (64, 65). A vast number of biochemical studies and the resolution of the crystallographic structures of maturation proteins, individually or in different complexes, showed that the maturation process is an intricate and dynamic pathway dependent on the formation of transient complexes of the Hyp proteins with each other (66, 67). The maturation proteins of [NiFe]-hydrogenases can be functionally separated into two groups according to the role they play. First, HypC, HypD, HypE, and HypF are responsible for synthesizing the $Fe(CN)_2CO$ moiety and inserting it into the precursor of the large subunit (68, 69). Subsequently, HypA and HypB are responsible for inserting the Ni ion in order to complete the bimetallic active site of the large subunit (65, 70). After the action of an endopeptidase that removes the C terminus of the immature large subunit (71), the latter can associate with the small subunit to build the catalytic unit of the [NiFe]-hydrogenase. The supplemental material provides a detailed dissection of individual maturation proteins.

**Maturation proteins of [NiFe]-hydrogenases in clostridial acetogens.** Based on literature data known to date, we were able to detect all six [NiFe]-hydrogenase maturation proteins in only three of eight *Clostridium* species, namely, *C. drakei*, *C. scatologenes*, and *C. carboxidivorans*, as recapitulated in Table 4. Our survey revealed that the HypB protein in these three species is a GTPase protein. It is interesting to note that we identified two putative homologs for each of the proteins HypC, HypD, and HypE in the genome of *C. carboxidivorans*, which deserves further investigation.

Our inspection of the *C. autoethanogenum*, *C. ljungdahlii*, *C. coskatii*, *C.* sp. AWRP, and *C. ragsdalei* genomes, using the conserved features reported in literature, did not allow us to reconstruct the complete maturation machinery of [NiFe]-hydrogenases since we detected the genes putatively encoding HypC, HypD, HypE, and HypF but not HypA and HypB.

**TABLE 2** Survey of transcript-level expression values for the hydrogenases predicted in *C. autoethanogenum*[a]

| Hydrogenase classification | Locus tag (RefSeq) | Locus tag (GenBank) | FPKM value | | | | RPKM value | | | |
|---|---|---|---|---|---|---|---|---|---|---|
| | | | Marcellin et al. (2016) [58] (35% CO, 10% $CO_2$, 2% $H_2$, 53% $N_2$) | Lemgruber et al. (2019) [59] (50% CO, 20% $CO_2$, 2% $H_2$, 28% Ar) | Lemgruber et al. (2019) [59] (50% CO, 20% $CO_2$, 20% $H_2$, 10% Ar) | Mock et al. (2015) [9] (42% CO, 20% $CO_2$, 2% $H_2$, 36% $N_2$) | Valgepea et al. (2017); low BC [60] (50% CO, 20% $CO_2$, 20% $H_2$, 10% Ar/$N_2$) | Valgepea et al. (2017); medium-low BC [60] (50% CO, 20% $CO_2$, 20% $H_2$, 10% Ar/$N_3$) | Valgepea et al. (2017); medium BC [60] (50% CO, 20% $CO_2$ 20% $H_2$, 10% Ar/$N_4$) | Valgepea et al. (2017); high BC [60] (50% CO, 20% $CO_2$, 20% $H_2$, 10% Ar/$N_5$) |
| NiFe 1a | CLAU_RS04060 | CAETHG_0861 | 5.01 | 1.5035 | 0.38 | <0.1 | 1.00 | 1.00 | 1.00 | 1.00 |
| | CLAU_RS04065 | CAETHG_0862 | 3.53 | 1.4325 | 0.73 | <0.1 | NA | NA | NA | NA |
| FeFe B | CLAU_RS00525 | CAETHG_0110 | 58.19 | 31.884 | 43.87 | 5.00 | 26.00 | 31.00 | 41.00 | 54.00 |
| FeFe A4 | CLAU_RS13700 | CAETHG_2798 | 6,130.87 | 5,040.1485 | 6,235.22 | 405.00 | 3,639.00 | 2,423.00 | 1,494.00 | 2,004.00 |
| | CLAU_RS13680 | CAETHG_2794 | 2,976.28 | 3,669.1405 | 4,436.60 | 312.00 | 2,558.00 | 1,604.00 | 853.00 | 1,214.00 |
| | CLAU_RS13685 | CAETHG_2795 | 4,073.53 | 3,117.7735 | 3,826.21 | 312.00 | 1,743.00 | 941.00 | 520.00 | 845.00 |
| | CLAU_RS13690 | CAETHG_2796 | 5,372.79 | 3,721.7915 | 4,575.29 | 312.00 | 2,559.00 | 1,405.00 | 798.00 | 1,438.00 |
| | CLAU_RS13695 | CAETHG_2797 | 7,373.13 | 4,108.917 | 5,225.86 | 405.00 | 2,610.00 | 1,612.00 | 968.00 | 1,383.00 |
| | CLAU_RS13705 | CAETHG_2799 | 32,618.50 | 5,787.024 | 6,488.30 | 405.00 | 3,861.00 | 3,066.00 | 1,953.00 | 2,398.00 |
| FeFe A3 | CLAU_RS17440 | CAETHG_3569 | 2.18 | 4.9865 | 4.92 | 2.00 | 4.00 | 4.00 | 6.00 | 7.00 |
| | CLAU_RS17445 | CAETHG_3570 | 0.67 | 1.666 | 1.76 | 2.00 | 1.00 | 1.00 | 2.00 | 2.00 |
| | CLAU_RS17450 | CAETHG_3571 | 0.76 | 0.459 | 0.33 | 2.00 | 0.00 | 0.00 | 0.00 | 0.00 |
| FeFe A4 | CLAU_RS18770 | CAETHG_3841 | 72.40 | 28.644 | 22.24 | 9.00 | 15.00 | 15.00 | 14.00 | 25.00 |
| | CLAU_RS18765 | CAETHG_3840 | 62.39 | 29.083 | 24.96 | NA | 14.00 | 13.00 | 11.00 | 23.00 |
| | CLAU_RS18775 | CAETHG_3842 | 8.87 | 17.231 | 15.61 | NA | 17.00 | 23.00 | 25.00 | 21.00 |

[a]The table displays the FPKM or RPKM values of individual hydrogenase subunits, reported in several RNA-seq analyses carried out in cell cultures grown on variously composed syngas (39–41). Column headers show syngas composition. NA reflects the cases where expression data are not available in the original reports. The study conducted in reference 41 considers different biomass concentrations (BCs). Correspondingly, the table reports the hydrogenases' expression levels for each BC: low BC (0.5 g dry cell weight [gDCW/L], medium-low BC (0.7 gDCW/L), medium BC (1.1 gDCW/L), and high BC (1.4 gDCW/L).

**TABLE 3** Overview of predicted [FeFe]-hydrogenase maturation proteins (HydE, HydG, HydF) in each acetogen of the genus *Clostridium*

| Microorganism | Gene encoding protein: | | |
| | HydE | HydG | HydF |
|---|---|---|---|
| *C. autoethanogenum* | CLAU_RS08145 | CLAU_RS03260 | CLAU_RS10040 |
| *C. carboxidivorans* | Ccar_RS19085 | | Ccar_RS05080 |
| *C. ljungdahlii* | CLJU_RS18880 | | CLJU_RS20880 |
| *C.* sp. AWRP | DMR38_RS19105 | | DMR38_RS21245 |
| *C. coskatii* | CLCOS_RS19995 | | CLCOS_RS18855 |
| *C. ragsdalei* | CLRAG_RS10265 | | CLRAG_RS02690 |
| *C. drakei* | B9W14_RS22750 | B9W14_RS10345 | B9W14_RS10005 |
| *C. scatologenes* | Csca_RS21720 | Csca_RS08575 | Csca_RS08935 |

**Diaphorase subunits in electron-bifurcating [FeFe]-hydrogenases.** Of the hydrogenases identified, the hexameric [FeFe]-hydrogenases of group A4 (9, 13) and the trimeric [FeFe]-hydrogenases of group A3 (72–74) can operate reversibly with a flavin-based electron-bifurcating mechanism using ferredoxin and NAD/NADP as electrons donor/acceptors. Indeed, when we analyzed the neighboring gene organization of such hydrogenases, we could identify a putative diaphorase subunit (as shown in Fig. 5 to 8). In the following section, we sought to gain some insights on the sequence/structural features involved in the cofactor binding.

Rossmann's introduction of the so-called Rossmann's fold (75, 76) was pioneering in the elucidation of the mode of interaction of enzymes with nucleotide-based cofactors. Albeit subject to a wide variability, the initial $\beta\alpha\beta$ fold of this structural motif is the most conserved segment among the different enzyme classes and the conserved glycine-rich (Gly-rich) loop between the first strand and the following helix (77).

As reported in several studies (18, 21, 78, 79), hydrogenase subunits and domains share sequence and structure similarities with other redox enzymes, such as complex I, also known as NADH:ubiquinone oxidoreductase (NUO), involved in the respiratory chains (78, 80–82). The noticed structural similarity suggests that the elucidation of the cofactor binding mode in hydrogenases could benefit from the transfer of knowledge acquired on the NAD binding mode of NuoF in complex I. The structural annotation of the putative diaphorase subunits in the identified hydrogenases showed domain similarity to the superfamily cluster NuoF (accession: COG1894; superfamily: cl34375) according to the Conserved Domain Database (CDD) (supplemental file 1). Thus, we investigated the sequence and structural features shared between our putative diaphorase subunits in group A4 and group A3 [FeFe]-hydrogenases and NuoF of complex I from *Thermus thermophilus*.

The structure of the entire complex I from *T. thermophilus* has been solved (83), and details about the interactions with flavin mononucleotide (FMN) and NADH in its NuoF subunit have been reported (84–86). Even though NuoF from *T. thermophilus* presents a Rossmann's fold domain (residues 73 to 240), the loop between the first strand and the following helix does not contain Gly residues, which at first suggested a different

**TABLE 4** Overview of predicted [NiFe]-hydrogenases maturation proteins (HypABCDEF) in each acetogen of the genus *Clostridium*

| Microorganism | HypA | HypB | HypC | HypD | HypE | HypF |
|---|---|---|---|---|---|---|
| *C. autoethanogenum* | | | CLAU_RS01740 | CLAU_RS01735 | CLAU_RS01730 | CLAU_RS01745 |
| *C. carboxidivorans* | Ccar_RS06395 | Ccar_RS06400 (GTP) | Ccar_RS06375, Ccar_RS09605 | Ccar_RS06370, Ccar_RS09600 | Ccar_RS06365, Ccar_RS09595 | Ccar_RS09610 |
| *C. ljungdahlii* | | | CLJU_RS11355 | CLJU_RS11350 | CLJU_RS11345 | CLJU_RS11360 |
| *C.* sp. AWRP | | | DMR38_RS11930 | DMR38_RS11925 | DMR38_RS11920 | DMR38_RS11935 |
| *C. coskatii* | | | CLCOS_RS19295 | CLCOS_RS19300 | CLCOS_RS19305 | CLCOS_RS19290 |
| *C. ragsdalei* | | | CLRAG_RS00845 | CLRAG_RS00850 | CLRAG_RS00855 | CLRAG_RS00840 |
| *C. drakei* | B9W14_RS11020 | B9W14_RS11025 (GTP) | B9W14_RS14125 | B9W14_RS14120 | B9W14_RS14115 | B9W14_RS14130 |
| *C. scatologenes* | Csca_RS07895 | Csca_RS07890 (GTP) | Csca_RS04330 | Csca_RS04335 | Csca_RS04340 | Csca_RS04325 |

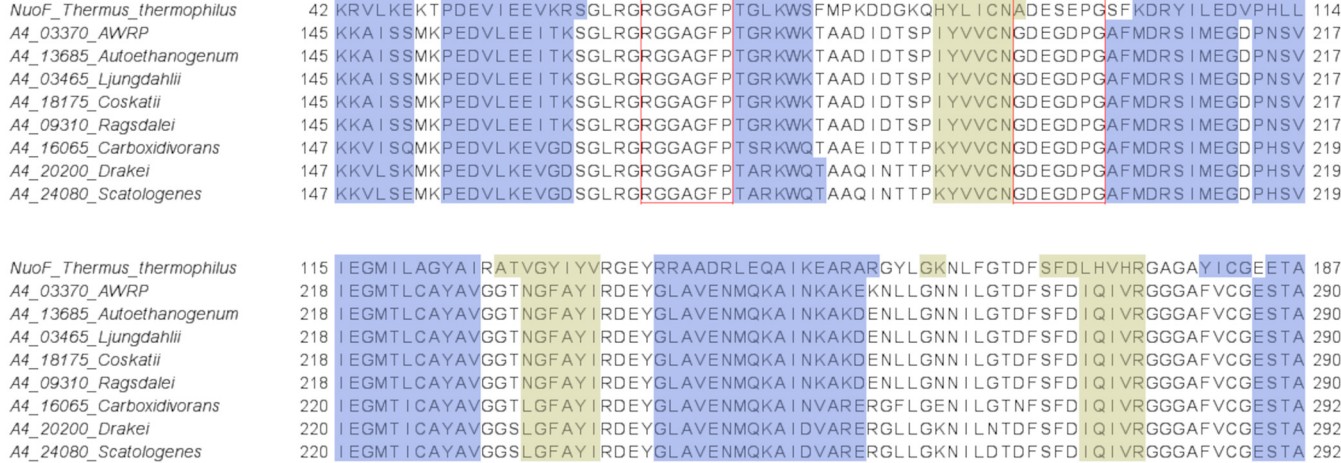

**FIG 9** Portion of the multiple sequence alignment of NuoF from *T. thermophilus* with the diaphorase subunits of the hexameric [FeFe]-hydrogenases of group A4. The figure focuses on the NuoF region in between the N-terminal domain and the Rossmann's fold domain. NuoF region involved in cofactor binding. Structural elements obtained by I-TASSER are color-coded. Red boxes: amino acids of the two conserved loops involved in the interaction with FMN and NAD/NADP; blue: α helices; yellow: β sheets.

mechanism of interaction with the cofactor of interest. In fact, the N-terminal domain (7 to 72) ends with a Gly-rich (R-G-G-A-G-F-P in *T. thermophilus*) loop (65 to 71), highly conserved in NuoF subunits (87, 88), which plays a key role in the interaction with NADH. It follows that in NuoF, an atypical Rossmann's fold domain, interacting with FMN, has evolved to accommodate both FMN and NADH in the binding pocket, with the cooperation of an additional Gly-rich loop at the N terminus (86, 89).

In the diaphorase subunits of the hydrogenases identified in our study, we confirmed the presence of the Gly-rich loop region at the end of the N-terminal domain of NuoF, with the conserved R-G-G-A-G-F-P motif in all group A4 [FeFe]-hydrogenases and the conserved R-G-G-G-G-F-P motif in all group A3 [FeFe]-hydrogenases (Fig. 9 and 10). Furthermore, for each examined subunit, the secondary structure prediction obtained by I-TASSER confirmed the presence of a secondary structure context typical of the Rossmann's fold domain downstream of the aforementioned Gly-rich loop (Fig. 9 and 10). However, different from NuoF, the loop connecting the first strand with the following helix in the Rossmann's fold domain is Gly rich. In particular, we could notice the occurrence of the G-D-E-G-D-P-G motif in all group A4 [FeFe]-hydrogenases. Thus, the analysis of the hydrogenases' diaphorase subunits allowed us to identify the presence of structural elements and arrangement thereof, which are characteristic of the NuoF cofactor-binding pocket (Fig. S3). Moreover, the *in silico* reconstruction with I-TASSER highlighted the existence of a region resembling the cofactor-binding pocket in NuoF (Fig. 11). However, the diaphorase subunits showed some distinctive features of their amino acid sequence in correspondence to the putative structural elements involved in the cofactor binding compared to those of NuoF.

The loop connecting the first strand with the following helix in the Rossman's fold domain of group A3 [FeFe]-hydrogenases is Gly rich, but, different from group A4 FeFe-hydrogenases, the conserved motif A-D-E-G-D-P-G features, except for a couple of cases, the replacement of the glycine in the first position with an alanine (Fig. S4). The exceptions are represented by the group A3 [FeFe]-hydrogenases of *C. drakei* (B9W14_RS20215 to B9W14_RS20225) and *C. scatologenes* (Csca_RS24050 to Csca_RS24060), where the first loop of the Rossmann's fold domain shares the same conserved G-D-E-G-D-P-G motif of the group A4 [FeFe]-hydrogenases. It is possible to hypothesize that the sequence motif corresponding to this loop could contribute to determining the cofactor binding specificity.

Since structural and spectroscopic determination is not available for the hydrogenases of the considered strains, we suggest that, to recognize the determinants of redox cofactor specificity, it may be useful to conduct targeted mutagenesis studies at

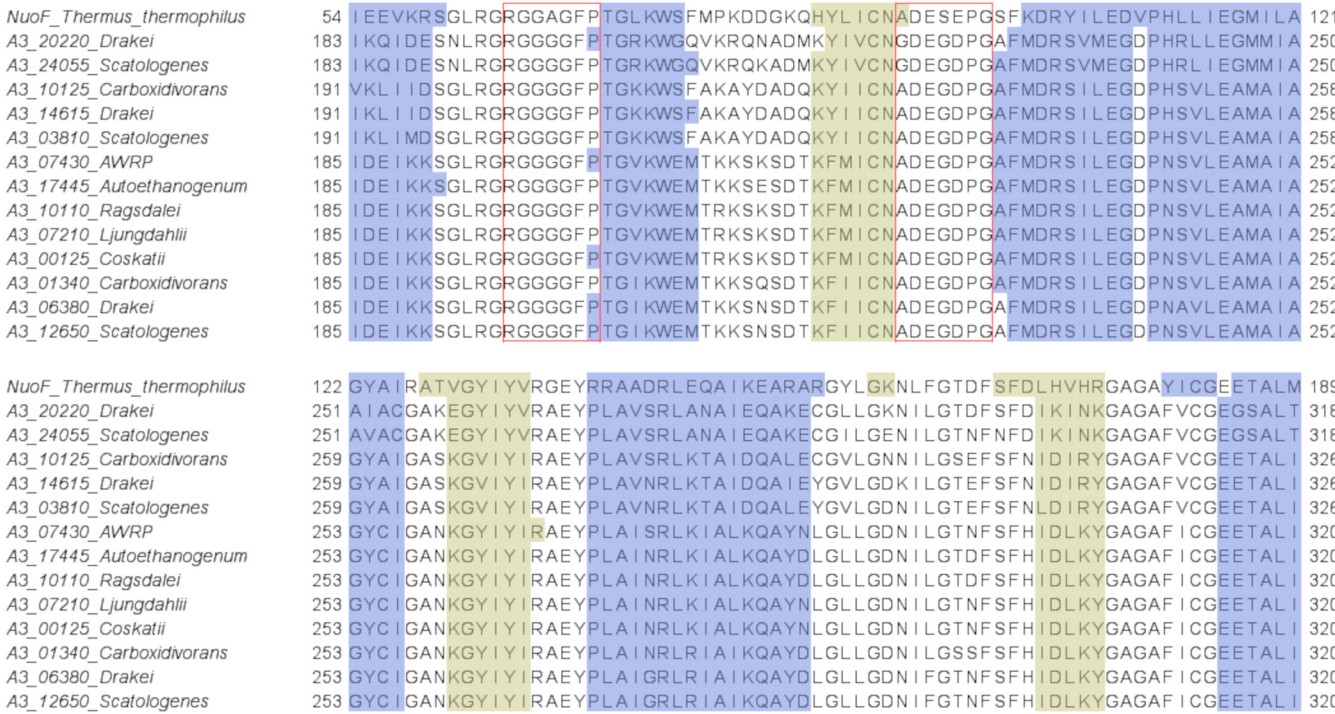

**FIG 10** Portion of the multiple sequence alignment of NuoF from *T. thermophilus* with the diaphorase subunits of the trimeric [FeFe]-hydrogenases of group A3. The figure focuses on the NuoF region in between the N-terminal domain and the Rossmann's fold domain. NuoF region involved in cofactor binding. Structural elements obtained by I-TASSER are color-coded. Red boxes: amino acids of the two conserved loops involved in the interaction with FMN and NAD/NADP; blue: α helices; yellow: β sheets.

the locations of the hydrogenase diaphorase subunit that map to the NuoF-like cofactor-binding pocket and show pronounced variability with respect to NuoF.

## DISCUSSION

In this study, we carried out a thorough characterization of the hydrogenases' repertoire in acetogens of the genus *Clostridium*, which, owing to the efficient carbon fixation pathway, are particularly appealing, with some of them already deployed in commercialized gas fermentation technology (4). To this aim, we relied on the availability of genome-scale amino acid sequence data to develop an entirely *in silico* approach which led us to the identification of candidate [NiFe]- and [FeFe]-hydrogenases and to their validation by the assessment of the existence of sequence-based patterns known to bind to the relevant metal clusters.

Each microorganism was found to encode a single H$_2$-uptake [NiFe]-hydrogenase enzyme belonging to group 1a, whose members primarily act as hydrogen oxidation catalysts, making hydrogen an additional usable inorganic electron donor for potentially generating energy (Fig. 3). Since H$_2$-uptake [NiFe]-hydrogenases are expectedly periplasmic, we searched the sequence motif usually recognized by the Tat pathway in the small subunits of these hydrogenases, but we found no supportive evidence. It is possible that the [NiFe]-hydrogenase small subunits could carry unusual signal peptides since naturally occurring Tat signal peptide variants have been previously reported mainly for *E. coli* (90–93). Alternatively, we cannot rule out the possibility that these hydrogenases resort to the general Sec protein transport (94, 95) or that their subcellular localization is cytoplasmic.

The [FeFe]-hydrogenase content of the acetogens investigated is composed of one group B [FeFe]-hydrogenase enzyme, of one to two group A4 [FeFe]-hydrogenases, and of one to three group A3 [FeFe]-hydrogenases (Fig. 3).

It was initially hypothesized that group B [FeFe]-hydrogenases could be involved in coupling reduced ferredoxin (Fd$_{red}$) oxidation to fermentative H$_2$ evolution (22) on the basis of

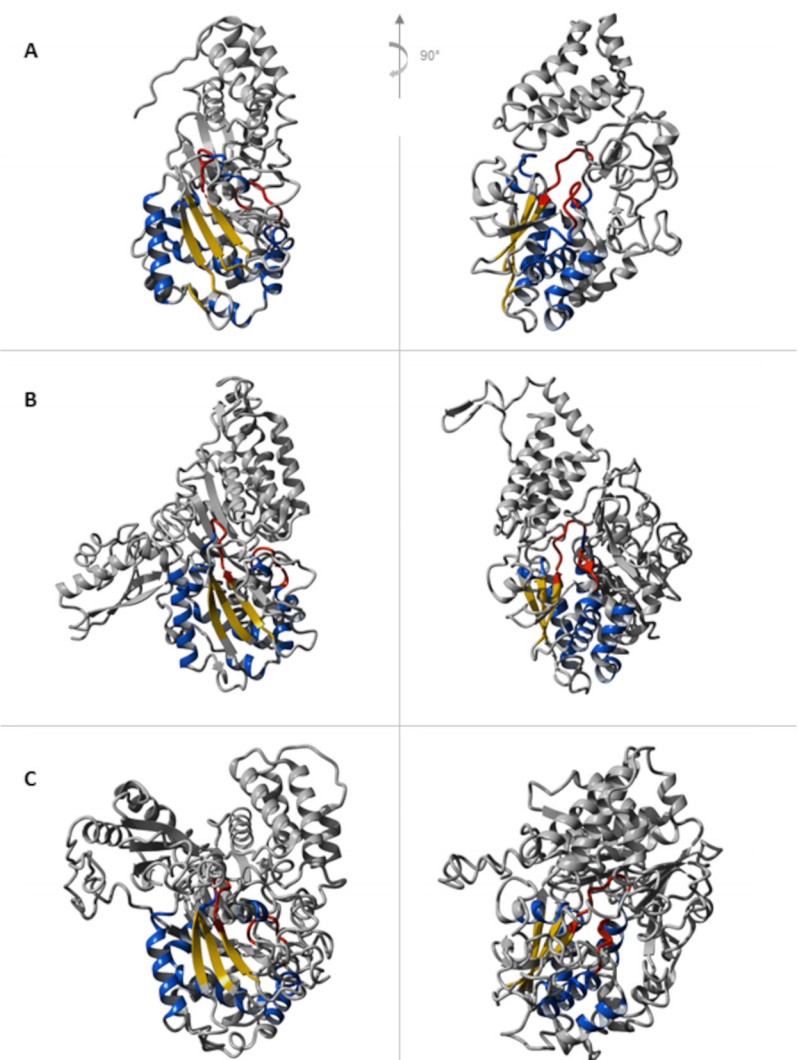

**FIG 11** Structure of NuoF compared to structures of representative group A4 and group A3 [FeFe]-hydrogenases computed by I-TASSER. (A) Structure of NuoF from *T. thermophilus* (PDB ID: 6ZIY). (B) I-TASSER 3D model predicted for the diaphorase subunit (CLAU_RS13685) of the hexameric hydrogenases encoded by CLAU_RS13680 to RS13705 in *C. autoethanogenum*. (C) I-TASSER 3D model predicted for the diaphorase subunit (CLAU_RS17445) of the trimeric hydrogenase encoded by CLAU_RS17440 to RS17450 in *C. autoethanogenum*. Red: conserved loops involved in the interaction with FMN and NAD/NADP; blue: $\alpha$ helices; yellow: $\beta$ sheets.

domain conservation and phylogenetic similarity with group A1 [FeFe]-hydrogenases. More recently, a biochemical and spectroscopic characterization of the [FeFe]-hydrogenase CpIII from *C. pasteurianum*, which belongs to group B2 as reported in reference 36, unveiled a distinctive H cluster population of catalytic intermediates compared to that of CpI (group A2) and CpII (group A3) [FeFe]-hydrogenases. More precisely, the study showed that CpIII features an extreme catalytic preference for $H_2$ production that was related to local differences in the H cluster environment (35).

Each acetogenic genome was found to contain one to three electron-bifurcating group A3 [FeFe]-hydrogenases.

Shortly after the discovery of flavin-based electron bifurcation (FBEB) (96, 97), a number of hydrogenases were shown to employ this mode of action to couple exergonic and endergonic oxidation–reduction reactions (98, 99) in order to fulfill various roles. The primary example of [FeFe]-hydrogenase that uses FBEB was isolated in *Thermotoga maritima*, where it utilizes the exergonic oxidation of ferredoxin to drive the unfavorable oxidation of NADH to produce $H_2$ (72). The convergence of electrons

from different sources of electron donor (NADH and $Fd_{red}$) to a single-electron acceptor (protons) is named electron confurcation (Fig. 4). Subsequently, the electron-bifurcating NAD- and ferredoxin-dependent [FeFe]-hydrogenases HydABCD and HydABC of the acetogens *Acetobacterium woodii* (100) and *Moorella thermoacetica* (73), respectively, and the NADP- and ferredoxin-dependent HydABC of *Thermoanaerobacter kivui* (74) were found able to operate in the electron confurcation mode to produce $H_2$ from organic substrates. The redox balancing achieved by electron confurcation increases the energy level in anaerobic microorganisms fermenting sugars since it allows acetyl coenzyme A (acetyl-CoA) to be converted to acetyl phosphate and then to acetate, yielding additional ATP by substrate-level phosphorylation (101).

FBEB in hydrogenases functioning in hydrogen-uptake mode allows the generation of reduced low-potential ferredoxin from comparably high-potential electron donors such as NADH or $H_2$. Reduced ferredoxin serves two purposes. It acts as the electron donor for reducing $CO_2$ to CO in the Wood-Ljungdahl pathway (102) and for anaerobic ion-motive respiratory chains leading to the synthesis of ATP (103–105). In this case, electron transfer from reduced ferredoxin to the electron acceptor leads to transport of ions across the cytoplasmic membrane and to the formation of a transmembrane electrochemical ion potential that drives the synthesis of ATP by an $F_1F_O$ ATP synthase (106).

It is worth noting that the aforementioned electron-bifurcating [FeFe]-hydrogenases serve the aim of both $H_2$ uptake and $H_2$ evolution, depending on the energetic needs of the acetogens when they grow in autotrophic or heterotrophic conditions. Elucidating the functions of the bifurcating group A3 [FeFe]-hydrogenases predicted in the acetogens considered in the present study is a daunting task that requires experimental assays of their biochemical and physiological properties.

Historically, group A4 [FeFe]-hydrogenases came in the spotlight thanks to the characterization of $HytABCDE_1E_2$ from *C. autoethanogenum* in complex with a formate dehydrogenase (9, 13), but the predicted formate dehydrogenase-linked [FeFe]-hydrogenases deserve experimental assessment in the remaining acetogens considered. According to the studies carried out in *C. autoethanogenum*, the complex catalyzes both the reversible coupled reduction of ferredoxin and NADP with $H_2$ or formate and the reversible formation of $H_2$ and $CO_2$ from formate. When CO is the electron donor, the reduction of 1 mol of $CO_2$ to formate is catalyzed by the redox-consuming formate dehydrogenase activity, which depends on 0.5 mol of NADPH and 0.5 mol of reduced ferredoxin (12). The function of the hydrogenase in the complex is puzzling and has been related to the protection of cells from overreduction of NADP and ferredoxin during growth on CO (13). During growth of *C. autoethanogenum* on $H_2$ and $CO_2$, $HytABCDE_1E_2$ appears to be active and provide the cells with electrons for reduction of ferredoxin and NADP as well as, in complex with formate dehydrogenase, for the reduction of $CO_2$ to formate (9). Likewise, gene expression data and metabolic flux balance analysis of *C. autoethanogenum* (9, 12), *C. ljungdahlii* (107), and *C. drakei* (19) grown on gas mixtures containing $H_2$ showed that the reduction of $CO_2$ to formate directly makes use of $H_2$ thanks to the formate-$H_2$ lyase activity of the hydrogenase/formate dehydrogenase enzyme complex.

In a previous study, hydrogenase inactivation using the ClosTron technology in *C. autoethanogenum* showed that the two group 4 [FeFe]-hydrogenases could not be disrupted and were concluded to be essential whereas the group B [FeFe]- and the group A3 [FeFe]-hydrogenases could be disrupted without a negative effect on growth in syngas (9). Noteworthily, when the group A4 [FeFe]-hydrogenases (CLAU_RS13680 to RS13705 and CLAU_RS18765 to RS18775) and the group B [FeFe]-hydrogenase (CLAU_RS00525) were targeted again in *C. autoethanogenum* LZ1561 (38) grown in CO-rich gas (44% CO, 32% $N_2$, 22% $CO_2$, and 2% $H_2$ or 50% CO, 21% $N_2$, 25% $CO_2$, and 4% $H_2$), they were found to be redundant. Indeed, the hydrogenase mutants appeared to grow, even though the mutant growth rates were different, the highest negative impact being recorded for mutants bearing inactivated group 4 [FeFe]-hydrogenase $HytABCDE_1E_2$ and the group B [FeFe]-hydrogenase. Interestingly, the scenario was different when the hydrogenase mutants grew on $H_2$-rich gas (65% $H_2$, 9.2% $N_2$, and 23%

$CO_2$). In this case, the mutant bearing inactivated Hyt was found to grow better than *C. autoethanogenum* LZ1561, indicating that other hydrogenases intervene to sustain growth, whereas the mutant bearing the group B [FeFe]-hydrogenase was found to grow poorly, which is potentially indicative of essentiality for survival in $H_2$-rich gas. In the light of the observed redundancy among hydrogenases which can bring about different gas uptake and carbon fixation profiles (38), it is of utmost relevance to characterize the $H_2$ uptake/evolving rates and cofactor dependency of the individual hydrogenases encoded by the genomes of the acetogens of the genus *Clostridium*.

**Maturation proteins of the identified [FeFe]- and [NiFe]-hydrogenases.** The bioinformatic identification of most of the maturation proteins for [NiFe]- and [FeFe]-hydrogenases corroborated the reliability of the inferred hydrogenases. Our bioinformatic analysis was articulated differently for [FeFe]- and [NiFe]-hydrogenases since their maturation relies on different sets of proteins.

The maturation of [FeFe]-hydrogenases involves the proteins HydE, HydF, and HydG. Whereas we detected putative HydE- and HydF-coding genes in each acetogen, we could detect putative HydG-coding genes only in *C. autoethanogenum*, *C. drakei*, and *C. scatologenes* (Table 3). We are not able to give a straightforward explanation for the missed identification of *hydG*, but we can hypothesize several scenarios.

First, we cannot exclude that the instances of the functional motifs at the C- and N-terminal ends of the HydG protein sequences do not thoroughly reflect in the motifs reported in the literature. The plausibility of this hypothesis is questionable in light of the fact that the acetogens under consideration are phylogenetically close to each other and the fact that we found that a few of these acetogens putatively encode an HydG protein (15, 108). Nonetheless, an analysis of the radical SAM protein sequences present in our genomes allowed us to identify, in all the acetogens except for *C. autoethanogenum*, a putative HydG-coding gene, which does not entirely fulfill the motifs typical of HydG according to the literature (Table S8). Several clues support this hypothesis. The putatively identified proteins have a similar amino acid length of known HydGs (~450 amino acids [aa]). Along with the presence of the canonical C-x(3)-C-x(2)-C motif of the radical SAM proteins, sequence analysis unveiled two conserved Cys-containing patterns in the C-terminal end. One of these Cys-containing patterns may coordinate an Fe-S cluster, which is known to be essential for the catalytic HydG activity, since the instances C-x(4)-C-x(23-25)-C, found in *C. carboxidivorans*, and C-x(4)-C-x(23-24)-C, found in the remaining acetogens, resemble the known motif C-x(2)-C-x(22)-C (see Fig. S2 in the supplemental material). It is worth noting that also *C. drakei* and *C. scatologenes* harbor a gene encoding a radical SAM protein and containing the same atypical Cys-based motif, C-x(4)-C-x(23-25)-C, as *C. carboxidivorans*. This observation might thus suggest that these two microorganisms also harbor a homologue gene of HydG. Further experimental studies are required in order to confirm or discard this hypothesis.

A second way to interpret the missed identification of *hydG* on the basis of known characteristic motifs originates from the *in vitro* demonstration that the activation of a [FeFe]-hydrogenase can occur coexpressing only HydE and HydF (HydF$^E$), without HydG, which should be the normal source of CO and CN$^-$ ligands (49). Whereas HydF$^E$ was capable of activating HydA$^{\Delta EFG}$ (*hydA* expressed in a genetic background devoid of the active site H cluster biosynthetic genes *hydE*, *hydF*, and *hydG*), albeit to a limited extent (only ~1% of that obtained by using all three the maturation proteins), HydF$^G$ did not provide any detectable activation of HydA. Notably, activation assays with HydF$^E$ showed that trace diatomics from the cellular environment could be incorporated into a [2Fe]-like precursor on HydF in the absence of HydG. Furthermore, supplementation of exogenous free CO or CN$^-$ to HydF$^E$ doubled the activation of HydA$^{\Delta EFG}$. It is interesting to note that exogenous CO and CN$^-$ were found to increase the ability of HydF$^{EG}$ to activate HydA$^{\Delta EFG}$ (53), indicating that also HydF$^{EG}$ harbors some partially assembled [2Fe] cluster precursors that, upon delivery of the dithiomethylamine ligand by HydF, can be activated by free diatomics from the cellular environment (53).

Following this course of reasoning, in the light of the toxicity of these compounds, we wonder if acetogens might have developed fine mechanisms, not yet identified to our knowledge, for the incorporation of ligands. This hypothesis may be also supported by the fact that HydG from *Clostridium acetobutylicum* has been demonstrated to produce CO and $CN^-$ in free form (49, 50, 52), which is indicative of the fact that the formation of the organometallic $Fe(CO)_2CN$ synthon is not essential for the incorporation of CO and $CN^-$ in the $[2Fe]_H$ cluster.

The maturation of [NiFe]-hydrogenases involves the proteins putatively encoding HypC, HypD, HypE, and HypF, for which we identified putative coding genes in all the acetogens considered, and HypA and HypB, for which we were able to identify putative coding genes only in *C. carboxidivorans*, *C. drakei*, and *C. scatologenes* (Table 4). It is known that, compared to the well-conserved biosynthesis, assembly, and insertion of the $Fe(CN)_2CO$ group by HypCDEF, the Ni-insertion process, conventionally mediated by HypA and HypB, is characterized by higher structural and functional diversity across different microorganisms (64, 66). This observation could suggest that the missed identification of putative HypA- and HypB-encoding genes in the aforementioned microorganisms, on the basis of currently known functional motifs, could be due to the usage of Ni traffic mechanisms which have not been reported yet in the literature for Ni insertion in [NiFe]-hydrogenases.

**Diaphorase subunits in the identified electron-bifurcating [FeFe]-hydrogenases.** Of the hydrogenases identified, the hexameric [FeFe]-hydrogenases of group A4 (9, 13) and the trimeric [FeFe]-hydrogenases of group A3 (72–74) can operate reversibly with a flavin-based electron-bifurcating mechanism using ferredoxin and NAD/NADP as electron donor/acceptors. Therefore, we inspected the sequence/structural features involved in the cofactor binding carried out by the diaphorase subunit identified in these hydrogenases (Fig. 5 to 8). To this aim, we relied on the previous observation that hydrogenase subunits and domains share sequence and structure similarities with other redox enzymes, such as complex I (78, 80–82). The structural similarity between the HydB subunit of the *T. maritima* electron-bifurcating [FeFe]-hydrogenase HydABC and the NuoF subunit of complex I (Birrell J, Furlan C, Chongdar N, Gupta P, Lubitz W, Ogata H, andBlaza J, unpublished data) was recently highlighted. Indeed, it has been hypothesized that the evolution of complex I could originate from the unification of prebuilt modules of hydrogenases and transporters (109, 110).

As a reference for the analysis of the hydrogenase subunits, we adopted complex I from *T. thermophilus* since the interaction between its NuoF subunit with flavin mononucleotide (FMN) and NADH has been investigated in several reports (84–86). It is worth noting that, in addition to the identification of structural elements characteristic of the NuoF cofactor-binding pocket, the diaphorase subunits of the hydrogenases showed some distinctive features of their amino acid sequence in correspondence to the putative structural elements involved in the cofactor binding. Dissimilar elements could be interpreted as pointing to the reliance on the binding to a cofactor different from NuoF. Indeed, the characterization of the electron-bifurcating [FeFe]-hydrogenase in *C. autoethanogenum* unveiled its NADP specificity (13). The loop connecting the first strand with the following helix in the Rossmann's fold in the diaphorase subunit of hydrogenases showed distinctive amino acid content compared to that of NuoF. Indeed, different from NuoF, the aforementioned loop in the group A4 [FeFe]-hydrogenases is glycine rich (G-D-E-G-D-P-G). Group A3 [FeFe]-hydrogenases feature a slightly different variation of the Gly-rich loop, where the first glycine is replaced by an alanine (A-D-E-G-D-P-G). We note just a couple of exceptions to the conservation of this in the group A3 [FeFe]-hydrogenases. Indeed, the first loop of the Rossmann's fold domain of the group A3 [FeFe]-hydrogenases of *C. drakei* (B9W14_RS20215 to B9W14_RS20225) and *C. scatologenes* (Csca_RS24050 to Csca_RS24060) shares the same conserved G-D-E-G-D-P-G motif of the group A4 [FeFe]-hydrogenases. It is possible to hypothesize that the sequence motif corresponding to this loop could contribute to determining the cofactor binding specificity. Indeed, the A-D-E-G-D-P-G motif occurs in the loop of the electron-bifurcating hydrogenase in *Moorella thermoacetica* and in *Desulfovibrio*

*fructosovorans*, which references 73 and 111 experimentally proved rely on NAD rather than NADP. On the other side, the G-D-E-G-D-P-G motif occurs in the loop of the *Thermoanaerobacter kivui* electron-bifurcating hydrogenase HydABC, which reference 74 showed depends on NADP but not NAD. Finally, the genes encoding the trimeric hydrogenases (B9W14_RS20215 to B9W14_RS20225, Csca_RS24050 to Csca_RS24060), which share the same motif in the loop of the Rossmann's fold domain with the group A4 [FeFe]-hydrogenases, are also genomically close to the genes coding the hexameric group A4 [FeFe]-hydrogenases, separated only by a formate dehydrogenase-coding gene. The exploration of potential interaction among such enzymes is worthy of consideration in light of the previously observed interaction of the hexameric group A4 [FeFe]-hydrogenases with the formate dehydrogenase (13).

In conclusion, our study provided detailed information on the hydrogenase composition of an unprecedented number of clostridial acetogens at the gene level. We identified hydrogenases from select acetogen genomes via a computational workflow, which involved comparison to a database of known hydrogenase sequences, with validation provided by reference to known catalytic sequence motifs and further analysis of neighboring genes. Albeit similar in terms of [NiFe]-hydrogenases, the acetogens considered showed increased variability in their [FeFe]-hydrogenases repertoire, with special reference to electron-bifurcating [FeFe]-hydrogenases. For the hydrogenases of the electron-bifurcating type, comparison to the diaphorase subunit of complex 1 identified several differences in structural motifs hypothesized to be relevant to NAD(P)H cofactor binding. The identified hydrogenases were further analyzed in the context of the maturation genes required for their biosynthesis, which revealed several interesting insights regarding absences of conserved genes in several acetogens. Our study represents a helpful resource to deepen our understanding of hydrogenases' functioning to develop future strain-engineering approaches and biotechnological approaches reliant on clostridial acetogens.

## MATERIALS AND METHODS

**Acetogen genomic data set.** Detailed information on the genomes of the clostridial acetogens (*C. autoethanogenum* DSM 10061, *C. ljungdahlii* DSM 13528, *C. carboxidivorans* P7, *C. drakei* SL1, *C. scatologenes* ATCC 25775, *C.* sp. AWRP, *C. ragsdalei* P11, and *C. coskatii* PTA-10522) is provided in the supplemental material.

**Reference data set of known hydrogenases.** A reference set of curated and nonredundant amino acid sequences corresponding to 3,265 hydrogenases was retrieved from the HydDB database (20). We then selected only [NiFe]-hydrogenase sequences and [FeFe]-hydrogenase sequences, which account for 2,012 and 1,228 sequences, respectively, along with their hydrogenase class, assigned according to the classification scheme developed within HydDB. This database contains the entire sequence of the [NiFe]-hydrogenase large subunits and only the H-domain of the [FeFe]-hydrogenase catalytic subunits to reduce the risk of misannotations due to the complex organization of [FeFe]-hydrogenases.

**Hydrogenase classification method.** In order to detect and functionally classify the hydrogenases, we devised the entirely computational approach depicted in Fig. 2.

**Inference of hydrogenase enzymes and hydrogenase class by alignment-based sequence analysis.** Protein sequences of each acetogen in the genus *Clostridium* under examination were aligned against the controlled catalogue of the [NiFe]- and [FeFe]-hydrogenases derived from HydDB by using the command-line application (112) of the blastp algorithm (113). Default parameters were used, except for the output format where the number 7 ("outfmt7") was chosen. To process the blastp output, we first sorted the hits in descending order of the following prioritized indices: percentage of sequence identity, E value, and query length. We subjected the sorted hits to a threshold on the E value (E value of $\leq 0.01$) and to a class-specific threshold on the percentage of sequence identity. The threshold on percent sequence identity varied according to the functional class assigned to the best hits of the query sequences. This classwise approach accounted for the fact that the intersequence identity of hydrogenase catalytic subunits varies when we consider different classes of hydrogenases annotated in HydDB (Table S1). Therefore, we retained a hit only if its percentage of identity against the query was higher than the minimum value of the distribution of the percentages of identity between the hydrogenase sequences of the class to which we predicted the hit belonged.

Of the hits that passed this filtering approach for each query, we selected the top four (except for some queries which were left with fewer than four hits after the thresholding) with which we predicted the class of the putative hydrogenase catalytic subunits identified in each acetogenic strain. Indeed, we assigned the predicted hydrogenase to the class (group or subgroup, if necessary) shared by the majority of the four top hits. In case of ties, we assigned the hydrogenase class of the top hit to the query sequence.

**Validation of putative hydrogenases by assessment of hydrogenase functional signatures.** We employed the following criteria to validate the reliability of the hydrogenase catalytic subunits predicted based on homology search. We considered a [FeFe]-hydrogenase catalytic subunit as validated if the protein sequence was found to contain an amino acid motif known to coordinate the Fe-S cluster of the H-domain. A [NiFe]-hydrogenase required the fulfillment of an additional criterion for validation purpose. Indeed, we explored the genes flanking each predicted [NiFe]-hydrogenase large catalytic subunit to seek the presence of a gene encoding a [NiFe]-hydrogenase small subunit. To identify the small subunit of a putative [NiFe]-hydrogenase, we carried out two checks.

First, the protein sequences flanking the predicted large catalytic subunits were retrieved and inputted to the InterPro protein families and domains database (114) and to the Conserved Domain Database (CDD) (115) in order to verify if the protein sequences were annotated as [NiFe]-hydrogenase small subunits.

Second, the protein sequences deemed as small subunits according to InterPro or CDD were searched for at least one instance of the amino acid patterns, which are known to ligate the Fe-S clusters in the small subunit of [NiFe]-hydrogenases that were identified in reference 39, using the ScanProsite tool (116). Previous surveys (21, 22, 117) on hydrogenase occurrences and biological functionality identified amino acid segments highly conserved around the Cys ligands of the metal clusters. The [NiFe]-hydrogenases feature the H cluster binding motifs referred to as L1 and L2, whereas the [FeFe]-hydrogenases feature the H cluster binding motifs referred to as P1, P2, and P3, which were initially recognized in reference 1 and subsequently updated in references 21, 22, and 117. Therefore, in this study, the catalogues of metal-binding motifs developed in references 21 and 22 were referenced to validate the identified hydrogenase catalytic subunits. In order to take into account the variability inherent to these patterns highlighted in reference 117, we opted for the usage of both sets of metal-binding motifs since they differ in granularity.

**Neighboring genetic organization.** As demonstrated in reference 22, genome architecture is a valuable hydrogenase classification tool. Therefore, to trace the possible structural composition of putative hydrogenases, we consulted the information on known hydrogenases reported on HydDB (20) (Data set S1). In particular, we investigated whether the number and type of subunits reported on HydDB for the hydrogenase class predicted by our analysis were traceable in the neighborhood of each candidate hydrogenase catalytic subunit. To this aim, we employed functional RefSeq and Uniprot annotations, as well as conserved domain annotations reported in UniParc and CDD, to verify the consistency of the neighboring genes of each catalytic subunit with the subunits expected in the hydrogenase class-specific template.

**Detection of putative genes encoding hydrogenase maturation proteins.** To verify the presence of [NiFe]- and [FeFe]-hydrogenase maturation genes, we performed a sequence-based alignment (blastp with standard parameters) of known maturation proteins against the amino acid sequences of each acetogen. As queries in the alignment-based analysis, we used the amino acid sequences of maturation protein-coding genes whose crystallographic structures are available in the Protein Data Bank (PDB). Since the framework of maturation pathway differs between [FeFe]- and [NiFe]-hydrogenases, the sets of query sequences differed between [NiFe]- and [FeFe]-hydrogenases. Regardless of the number of hits obtained for each query sequence, we then inspected each hit, searching for the conserved amino acids signatures reported by the literature to date for the recognition of hydrogenase maturation proteins. Full details are available in the supplemental material.

**Diaphorase subunits of group A4 and group A3 [FeFe]-hydrogenases.** We analyzed the protein domain annotations of the subunits of each validated [FeFe]-hydrogenase of the groups A3 and A4 by interrogating CDD (115) with the CD-search tool (118) in order to check for the existence of a subunit that features structural domains typical of the complex I NuoF subunit (accession: COG1894; superfamily: cl34375). Upon positive outcome of this verification, we assessed if the genomic position of the putative cofactor-binding subunit matches up with the neighboring genomic organization typical of the predicted hydrogenase class (20).

To add a further level of detail, we compared the sequences and the structural features of these subunits with the NuoF subunit of *T. thermophilus* (PDB ID: 6ZIY), as detailed in the supplemental material.

## SUPPLEMENTAL MATERIAL

Supplemental material is available online only.
**SUPPLEMENTAL FILE 1**, PDF file, 2.8 MB.
**SUPPLEMENTAL FILE 2**, XLSX file, 0.5 MB.

## ACKNOWLEDGMENTS

We thank Chris Greening at Monash University's Biomedicine Discovery Institute for useful suggestions and the reviewers for their thoughtful comments and efforts toward improving the manuscript.

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
