## [Reviewer comments · Microbiology Spectrum]

Microbiology Spectrum

Genome-scale mining of acetogens of the genus *Clostridium* unveils distinctive traits in [FeFe]- and [NiFe]- hydrogenase content and maturation

Pier Francesco Di Leonardo, Giacomo Antonicelli, Valeria Agostino, and Angela Re

Corresponding Author(s): Angela Re, Istituto Italiano di Tecnologia

Review Timeline:

Submission Date:	March 20, 2022
Editorial Decision:	May 4, 2022
Revision Received:	June 2, 2022
Accepted:	June 5, 2022

Editor: Emily Weinert

Reviewer(s): The reviewers have opted to remain anonymous.

Transaction Report:

DOI: <https://doi.org/10.1128/spectrum.01019-22>

May 4, 2022

Dr. Angela Re
Istituto Italiano di Tecnologia
Centre for Sustainable Future Technologies
Via Livorno 60
Torino, Piemonte 10144
Italy

Re: Spectrum01019-22 (Genome-scale mining of acetogens of the genus *Clostridium* unveils distinctive traits in [FeFe]-hydrogenase content and maturation)

Dear Dr. Angela Re:

Thank you for submitting your manuscript to Microbiology Spectrum. When submitting the revised version of your paper, please provide (1) point-by-point responses to the issues raised by the reviewers as file type "Response to Reviewers," not in your cover letter, and (2) a PDF file that indicates the changes from the original submission (by highlighting or underlining the changes) as file type "Marked Up Manuscript - For Review Only". Please use this link to submit your revised manuscript - we strongly recommend that you submit your paper within the next 60 days or reach out to me. Detailed instructions on submitting your revised paper are below. The major changes requested are organizational; currently the manuscript is difficult to follow for a non-expert in the hydrogenase field. Separating the Results and Discussions sections so that the new data are clearly outlined in the Results section and the implications of your data on the field are outlined in the Discussion section may be helpful.

Link Not Available

Sincerely,

Emily Weinert

Journals Department
Reviewer comments:

Reviewer #1 (Comments for the Author):

The manuscript by Di Leonardo et al provides an in-depth genomic analysis of clostridial acetogen type [FeFe]- and [NiFe]-hydrogenases. The approach was to identify hydrogenases from select acetogen genomes and validate via a computational workflow which involved comparison to a database of hydrogenase sequences, with further validation provided by reference to known catalytic sequence motifs and analysis of neighboring genes. The results were further analyzed in context of their maturation genes required for biosynthesis, which revealed several interesting insights regarding absences of conserved genes

in several cases. For the electron-bifurcating type hydrogenases identified, comparison to the diaphorase subunit of complex 1 identified several differences in structural motifs hypothesize to be relevant to NAD(P)H cofactor binding.

Overall, I found that the manuscript fills an important knowledge gap in hydrogenase literature as hydrogenases from clostridial acetogens are less studied. The manuscript is well referenced and provides a helpful interpretation of the genomic results considering previous experimental work and provides basis for future characterization of the hydrogenases identified. Several aspects of the manuscript could be improved to increase accessibility to the general audience and to strengthen the impact of the work. Notably missing is a conclusions section, which should be added to summarize the primary findings and relevance to the field.

Specific comments: The title doesn't adequately reflect the content, as the title is specific for [FeFe]-hydrogenase but the work informs on [NiFe]- and [FeFe]-hydrogenase. For the introduction (paragraph starting on line 83), it would be helpful to incorporate a figure depicting relevant acetogen energetic pathways with proposed functional roles of [FeFe]- and [NiFe]-hydrogenases. On lines 78-80, the statement that "H₂ evolution is carried out by [FeFe]- or [NiFe]-hydrogenases while H₂ oxidation is typically but not exclusively, attributed to [NiFe]-hydrogenases" is somewhat confusing as there are many examples of [FeFe]-hydrogenases that show preference for H₂ oxidation over H₂ evolution. Line 83, there is a typo in "through." Figure legends should be added for Figures 1 and 2. I found Figure 1 a bit hard to follow with the different abbreviations and these could be spelled out in the legend. In the results section starting on line 123, "groups" of hydrogenases are referred to, however the terminology is not properly introduced. I gather this is taking on the classification system by Calusinska et al in Microbiology 2010, 156, 1575-1588 (?) A few sentences about how [NiFe]- and [FeFe]- are classified into different groups and what these are would be helpful before reporting the results. Line 176 states that no representative [FeFe]-hydrogenase of subgroup B has been biochemically characterized. It should be noted that in Artz et al., J. Am. Chem. Soc. 2020, 142, 1227 there is initial biochemical and spectroscopic characterization of CpIII [FeFe]-hydrogenase from *C. pasterianum* (group B2 as reported by Therien et al. Frontiers Microbiology, 2017, 1305) which show an altered activity profile and spectroscopic properties compared to Cpl (Group A2) and CplI (Group A3) [FeFe]-hydrogenases. For the paragraph starting on Line 178, a figure or schematic depicting how electron-bifurcating (or confurcating) [FeFe]-hydrogenase function in redox pathways via H₂ uptake or evolution directions would be helpful. Line 456 (and 636), would recommend modifying the title "Redox cofactor binding subunits" to something that includes diaphorase since redox cofactor could be interpreted as something else. For Figures 7 and 8, would recommend incorporating a structural depiction of NuoF showing cofactor binding regions and conserved loops, using the coloring scheme of sequence alignment. The structural results with I-Tasser should also be brought into Figure 7 so any differences in the cofactor binding region of the hydrogenase diaphorase compared to NuoF can be visualized.

Reviewer #2 (Comments for the Author):

Dear authors,

I found your manuscript on hydrogenases rather intriguing because, as you mentioned in your introduction, they are widespread and fulfill diverse functions in microorganisms.

Unfortunately I found the manuscript rather challenging to read, as there is no clear flow of thought throughout the manuscript. Much of the results and discussion part reads more like a review article, and it is difficult to identify what role your own results play. It reads more like a list of all the facts that you could find in the literature than a well conducted discussion of your own results. This makes your results and discussion part way too long and you lose the reader along the way. Looking at your data in the tables and figures, you seem to have done some quite solid work, so you should really try to emphasize what you have done and only represent that part of the literature that is clearly needed in the discussion of your data. I would suggest that you separate the "results and discussion" into two separate chapters, "results" and "discussion", in order to better focus on your own results.

Staff Comments:

Preparing Revision Guidelines

- Point-by-point responses to the issues raised by the reviewers in a file named "Response to Reviewers," NOT IN YOUR COVER LETTER.
- Upload a compare copy of the manuscript (without figures) as a "Marked-Up Manuscript" file.
- Each figure must be uploaded as a separate file, and any multipanel figures must be assembled into one file.
- Manuscript: A .DOC version of the revised manuscript

- Figures: Editable, high-resolution, individual figure files are required at revision, TIFF or EPS files are preferred

Please return the manuscript within 60 days; if you cannot complete the modification within this time period, please contact me. If you do not wish to modify the manuscript and prefer to submit it to another journal, please notify me of your decision immediately so that the manuscript may be formally withdrawn from consideration by Microbiology Spectrum.

We provide a point-by-point responses to the issues raised by the Reviewers. The comments of the Reviewers are in blue, the responses of the Authors are reported in black and the changes applied to the manuscript (when transferrable directly in the present document) are reported in red.

Reviewer #1 (Comments for the Author):

The manuscript by Di Leonardo et al provides an in-depth genomic analysis of clostridial acetogen type [FeFe]- and [NiFe]- hydrogenases. The approach was to identify hydrogenases from select acetogen genomes and validate via a computational workflow which involved comparison to a database of hydrogenase sequences, with further validation provided by reference to known catalytic sequence motifs and analysis of neighboring genes. The results were further analyzed in context of their maturation genes required for biosynthesis, which revealed several interesting insights regarding absences of conserved genes in several cases. For the electron-bifurcating type hydrogenases identified, comparison to the diaphorase subunit of complex 1 identified several differences in structural motifs hypothesize to be relevant to NAD(P)H cofactor binding.

Overall, I found that the manuscript fills an important knowledge gap in hydrogenase literature as hydrogenases from clostridial acetogens are less studied. The manuscript is well referenced and provides a helpful interpretation of the genomic results considering previous experimental work and provides basis for future characterization of the hydrogenases identified. Several aspects of the manuscript could be improved to increase accessibility to the general audience and to strengthen the impact of the work. Notably missing is a conclusions section, which should be added to summarize the primary findings and relevance to the field.

We thank the Reviewer for his/her comments. Even though we did not created a dedicated Conclusion section, we restructured the manuscript to enhance the ability to discern our findings from the context of the literature that was used to interpret our results, in line with the request formulated by the Reviewer #2. Within this attempt, we separated the Results section from the Discussion one and, at the end of the latter, we included a brief summary of our results. We modified the manuscript as follows:

“In conclusion, our study provided detailed information on the hydrogenase composition of an unprecedented number of clostridial acetogens at the gene level. We identified hydrogenases from select acetogen genomes via a computational workflow, which

involved comparison to a database of known hydrogenase sequences, with validation provided by reference to known catalytic sequence motifs and further analysis of neighboring genes. Albeit similar in terms of [NiFe]-hydrogenases, the acetogens considered showed increased variability in their [FeFe]-hydrogenases repertoire, with special reference to electron-bifurcating [FeFe]-hydrogenases. For the hydrogenases of the electron-bifurcating type, comparison to the diaphorase subunit of Complex 1 identified several differences in structural motifs hypothesized to be relevant to NAD(P)H cofactor binding. The identified hydrogenases were further analyzed in the context of the maturation genes required for their biosynthesis, which revealed several interesting insights regarding absences of conserved genes in several acetogens. Our study represents a helpful resource to deepen our understanding of hydrogenases' functioning, to develop future strain engineering approaches and biotechnological approaches reliant on clostridial acetogens.”

Specific comments:

The title doesn't adequately reflect the content, as the title is specific for [FeFe]-hydrogenase but the work informs on [NiFe]- and [FeFe]-hydrogenase.

We agree with the Reviewer and we modified the title as follows: “Genome-scale mining of acetogens of the genus *Clostridium* unveils distinctive traits in [FeFe]- and [NiFe]-hydrogenase content and maturation”

For the introduction (paragraph starting on line 83), it would be helpful to incorporate a figure depicting relevant acetogen energetic pathways with proposed functional roles of [FeFe]- and [NiFe]-hydrogenases.

We thank the Reviewer for asking for this supplement of information that we integrated in the Figure 1 of the revised version of the manuscript.

Figure 1. Schematic overview of energy-conserving mechanisms in acetogens of the genus

Clostridium without providing exact stoichiometries. (A) With H₂ as electron source, the reducing equivalents for the reductive steps in the Wood-Ljungdahl pathway are provided by an H₂-oxidizing, electron-bifurcating hydrogenase/formate dehydrogenase complex (HytA-E/FDH) which reduces Fd, NADP and CO₂. **(B)** With CO as electron source, the reducing equivalents for the reductive steps are provided by the CO dehydrogenase/acetyl coenzyme A synthase (CODH/ACS) which reduces Fd. The hydrogenase protects the cells from over-reduction when NADP and ferredoxin get too reduced during growth on CO.

The electron-bifurcating and ferredoxin-dependent transhydrogenase Nfn is transferring electrons between Fd, NADH and NADPH. The methylene-THF reductase is assumed to be electron bifurcating. Excess reduced ferredoxin (Fd_{red}) is oxidized by the Rnf complex, which reduces NAD and builds up an H⁺ gradient. This gradient drives ATP synthesis via the H⁺-dependent ATP synthase.

Cofactors and energy equivalents are coded, respectively, in blue and green colors. Bold arrows denote by-products' exchange reactions. Dashed arrowed lines denote reactions' sets that were collapsed for ease of reading. Metabolites are displayed in the figure. THF stands for tetrahydrofolate. Capital letters in red denote enzymes. FTHFL, Formate:tetrahydrofolate ligase; MTHFC, Methenyltetrahydrofolate cyclohydrolase; MTHFD, Methylene-tetrahydrofolate dehydrogenase; MTHFR, electron bifurcating; NAD-dependent electron-bifurcating methylene-tetrahydrofolate reductase, METR, Methyltetrahydrofolate:corrinoid methyltransferase;

ACALDx/y, acetaldehyde:NAD(P) oxidoreductase; ALCDx/y, Ethanol:NAD(P) oxidoreductase; PTA, Phosphate acetyltransferase; ACK, Acetate kinase; AOR, Acetaldehyde:ferredoxin oxidoreductase.

On lines 78-80, the statement that "H₂ evolution is carried out by [FeFe]- or [NiFe]-hydrogenases while H₂ oxidation is typically but not exclusively, attributed to [NiFe]-hydrogenases" is somewhat confusing as there are many examples of [FeFe]-hydrogenases that show preference for H₂ oxidation over H₂ evolution.

We thank the Reviewer for pointing out this confusing sentence that we removed from the revised version of the manuscript.

Line 83, there is a typo in "through."

We amended the manuscript correspondingly.

Figure legends should be added for Figures 1 and 2. I found Figure 1 a bit hard to follow with the different abbreviations and these could be spelled out in the legend.

We thank the Reviewer for highlighting this missing piece of information in relation to Figure 1 (presently Figure 2) and Figure 2 (presently Figure 3), that we added in the revised version of the manuscript. Since previous Figure 1 (presently Figure 2) was hard to follow, we sought to enhance its understanding, with particular reference to limiting the usage of abbreviations and restyling specific sections such as step 3 and step 4. We hope that both figures, coupled with their own legends, have been improved.

The legend of Figure 3 was modified as follows.

Figure 3. Distribution of predicted [NiFe]- and [FeFe]-hydrogenase catalytic subunits in the acetogens of the genus *Clostridium* considered in this study. The barplot displays the number of hydrogenases of each subgroup in each acetogen of the genus *Clostridium* examined in this study. Hydrogenase subgroups are color-coded.

Figure 2 was modified as follows:

Figure 2. Computational pipeline for the identification and validation of hydrogenase catalytic subunits. The computational workflow consists of several steps. **1. Alignment:** Protein sequences of each acetogen were aligned against the controlled catalogue of [NiFe]- and [FeFe]-hydrogenases derived from HydDB. We sorted the hits in descending order of the following prioritized indices: percentage of sequence identity, E-value and query length. **2. Hit selection:** sorted hits were retained if the corresponding E-value ≤ 0.01 and if its percentage of sequence identity against the query met a class-specific criterion. Given the predicted class of the hit of a certain query sequence, we computed the distribution of the percentages of sequence identity between the hydrogenase sequences belonging to that class in HydDB. By way of example, we depicted the distributions corresponding to different classes in different colors in the figure. We retained an alignment hit if its percentage of identity against the query was higher than the minimum value of the aforementioned distribution. **3. Hydrogenase class assignment:** we assigned the predicted hydrogenase to the class (group or subgroup, if necessary) shared by the majority of the four top hits. **4. Hydrogenase validation:** We deemed a [FeFe]-hydrogenase catalytic subunit as validated if the protein sequence contained an amino acid motif known to coordinate the Fe-S cluster of the H-domain. For validating a [NiFe]-hydrogenase we also explored the genes flanking each predicted [NiFe]-hydrogenase large catalytic subunit to ascertain the presence of a gene encoding a [NiFe]-hydrogenase small subunit. Abbreviations: H₂ase, hydrogenase; Seq. Ident., sequence identity; E-value, expected value; a. a.: amino acid.

In the results section starting on line 123, "groups" of hydrogenases are referred to, however the terminology is not properly introduced. I gather this is taking on the classification system by Calusinska et al in Microbiology 2010, 156, 1575-1588 (?) A few sentences about how [NiFe]- and [FeFe]- are classified into different groups and what these are would be helpful before reporting the results.

We thank the Reviewer for pointing out this aspect. We amended it by introducing the classification scheme in the Results section (lines 116-125 of the compare copy of the manuscript) as follows:

“To this aim, we relied on the hydrogenase classification scheme predictive of biological functionality available at HydDB (20), which descends from the definition provided in refs. (1, 18, 21, 22). Indeed, [FeFe]- and [NiFe]-hydrogenases are hierarchically classified into different groups and subgroups that differ from each other by biochemical features and functional role such as respiration, sensing, fermentation. The HydDB classification scheme is primarily based on the topology of phylogenetic trees built from the amino acid sequence of hydrogenase catalytic subunits/domains, and it includes 29 subgroups (within four groups) of [NiFe]-hydrogenases and six subgroups (within three groups) of [FeFe]-hydrogenases.”

Line 176 states that no representative [FeFe]-hydrogenase of subgroup B has been biochemically characterized. It should be noted that in Artz et al., J. Am. Chem. Soc. 2020, 142, 1227 there is initial biochemical and spectroscopic characterization of CpIII [FeFe]-hydrogenase from *C. pasteurianum* (group B2 as reported by Therien et al. Frontiers Microbiology, 2017, 1305) which show an altered activity profile and spectroscopic properties compared to Cpl (Group A2) and CplI (Group A3) [FeFe]-hydrogenases.

We thank the Reviewer for noting this mistake. The revised version of the manuscript now includes this information in the Results section “[FeFe]-hydrogenases in clostridial acetogens” (lines 188-194 of the compare copy of the manuscript). Indeed, the Results section of the manuscript was modified as follows:

*“A representative hydrogenase of this subgroup, CpIII in *C. pasteurianum*, has been recently biochemically and spectroscopically characterized in comparison to the group A2*

Cpl and to the group A3 CplI [FeFe]-hydrogenases (35, 36). Preferential stabilization of key catalytic intermediates through subtle changes in the outer coordination sphere was found to result in stabilization and/or destabilization of different oxidation states. As a result of this characterization, CplII was found to have a catalytic bias towards H₂ production (35, 36).”

The Discussion section was modified as follows:

*“It was initially hypothesized that Group B [FeFe]-hydrogenases could be involved in coupling reduced ferredoxin (Fd_{red}) oxidation to fermentative H₂ evolution (22) on the basis of domain conservation and phylogenetic similarity with group A1 [FeFe]-hydrogenases. More recently, a biochemical and spectroscopic characterization of the [FeFe]-hydrogenase CplII from *C. pasterianum*, which belongs to group B2 as reported in (36), unveiled a distinctive H cluster population of catalytic intermediates compared to CplI (Group A2) and CplII (Group A3) [FeFe]-hydrogenases. More precisely, the study showed that CplII features an extreme catalytic preference for H₂ production that was related to local differences in the H cluster environment (35).”*

For the paragraph starting on Line 178, a figure or schematic depicting how electron-bifurcating (or confurcating) [FeFe]-hydrogenase function in redox pathways via H₂ uptake or evolution directions would be helpful.

We agree with the Reviewer and the current version of the manuscript now includes Figure 4 to address this request.

Bifurcating/conforcating hydrogenases. (A) Flavin-based electron bifurcating hydrogenase oxidizes an electron donor (H₂) and delivers the electrons simultaneously to two different electron acceptors. The reduction of a high-potential acceptor (NAD⁺) is exergonic and drives the endergonic reduction of a low-potential acceptor (Ferredoxin). (B) The flavin-based electron conforcating hydrogenase oxidizes simultaneously NADH and ferredoxin to produce H₂. In this scenario, the electron pairs from the two electron donors, NADH and reduced ferredoxin, converge to reduce protons to H₂. Flavin is the cofactor family with bifurcating/conforcating property. (C) Conceptual illustration of ideal electron bifurcation whereby the transfer of electrons of intermediate reduction potential to a bifurcating site (flavin) is parsed out to acceptors with, respectively, more positive and more negative reduction potential but whose sum is equivalent to the overall reduction potential of the donated electrons. (D) A two-electron acceptor of intermediate reduction potential simultaneously accepts electrons from electron donors with, respectively, more negative and more positive reduction potentials.

Line 456 (and 636), would recommend modifying the title "Redox cofactor binding subunits" to something that includes diaphorase since redox cofactor could be interpreted as something else.

We thank the Reviewer for this suggestion and we adopted the following title: "Diaphorase subunits in electron-bifurcating [FeFe]-hydrogenases".

For Figures 7 and 8, would recommend incorporating a structural depiction of NuoF showing cofactor binding regions and conserved loops, using the coloring scheme of sequence alignment. The structural results with I-

Tasser should also be brought into Figure 7 so any differences in the cofactor binding region of the hydrogenase diaphorase compared to NuoF can be visualized.

We agree with the Reviewer and we included Figure 11 to compare the structure of NuoF to the structures of the diaphorase subunits of, respectively, a representative group A4 [FeFe]-hydrogenase and of a representative group A3 [FeFe]-hydrogenase, encoded in the *C. autoethanogenum* genome. The structures of the representative subunits were obtained through I-TASSER. The structural elements relevant for the discussion of the cofactor binding mode are color-coded in agreement with the color code used in Figures 9-10 (previously Figures 7-8) showing the sequence alignment of NuoF from *T. thermophilus* with the cofactor binding subunits of the hexameric [FeFe]-hydrogenases of group A4 and the trimeric [FeFe]-hydrogenases of group A3.

Figure 11. Structure of NuoF compared to structures of representative group A4 and group A3 [FeFe]-hydrogenases computed by I-TASSER. (A) Structure of NuoF from *T. thermophilus* (PDB ID: 6Z1Y). **(B)** I-TASSER 3D model predicted for the diaphorase subunit (CLAU_RS13685) of the hexameric hydrogenases encoded by CLAU_RS13680-RS13705 in *C. autoethanogenum*. **(C)** I-TASSER 3D model predicted for the diaphorase subunit (CLAU_RS17445) of the trimeric hydrogenase encoded by CLAU_RS17440-RS17450 in *C. autoethanogenum*. Red: conserved loops involved in the interaction with

FMN and NAD/NADP; blue: α helices; yellow: β sheets.

Reviewer #2 (Comments for the Author):

Dear authors,

I found your manuscript on hydrogenases rather intriguing because, as you mentioned in your introduction, they are widespread and fulfill diverse functions in microorganisms.

Unfortunately I found the manuscript rather challenging to read, as there is no clear flow of thought throughout the manuscript. Much of the results and discussion part reads more like a review article, and it is difficult to identify what role your own results play. It reads more like a list of all the facts that you could find in the literature than a well conducted discussion of your own results. This makes your results and discussion part way too long and you lose the reader along the way. Looking at your data in the tables and figures, you seem to have done some quite solid work, so you should really try to emphasize what you have done and only represent that part of the literature that is clearly needed in the discussion of your data. I would suggest that you separate the "results and discussion" into two separate chapters, "results" and "discussion", in order to better focus on your own results.

We thank the Reviewer for his/her comments. In line with the suggestions, we restructured the manuscript in order to highlight the findings of our work. To this aim, we created separated sections for presenting the results and for their discussion in the context of the literature. We believe that, thanks to this change, the manuscript has improved in its readability.

June 5, 2022

Dr. Angela Re
Istituto Italiano di Tecnologia
Centre for Sustainable Future Technologies
Via Livorno 60
Torino, Piemonte 10144
Italy

Re: Spectrum01019-22R1 (Genome-scale mining of acetogens of the genus *Clostridium* unveils distinctive traits in [FeFe]- and [NiFe]-hydrogenase content and maturation)

Dear Dr. Re:

Thank you for your careful modification of your manuscript to address the reviewer comments. Your manuscript has been accepted, and I am forwarding it to the ASM Journals Department for publication. You will be notified when your proofs are ready to be viewed.

Sincerely,

Emily Weinert
Editor, Microbiology Spectrum

Journals Department
Supplemental Dataset: Accept
Supplemental Material: Accept